**Technical Report**

# High-throughput peptide-centric local stability assay extends protein–ligand identification to membrane proteins, tissues and bacteria

Kejia Li [1,4], Clement M. Potel [1,4], Isabelle Becher[1], Nico Hüttmann [1], Martin Garrido-Rodriguez[1,2], Jennifer Schwarz [3], Mira Lea Burtscher[1] & Mikhail M. Savitski [1] ✉

Systematic mapping of protein–ligand interactions is essential for understanding biological processes and drug mechanisms. Peptide-centric local stability assay (PELSA) is a powerful tool for detecting these interactions and identifying potential binding sites. However, its original workflow is limited in throughput, sample compatibility and accessible protein targets. Here, we introduce a high-throughput adaptation—HT-PELSA—that increases sample processing efficiency by 100-fold while maintaining high sensitivity and reproducibility. HT-PELSA substantially extends the capabilities of the original method by enabling sensitive protein–ligand profiling in crude cell, tissue and bacterial lysates, allowing the identification of membrane protein targets in diverse biological systems. We demonstrate that HT-PELSA can precisely and accurately determine binding affinities of small molecule inhibitors, sensitively detect direct and allosteric ATP binding sites, and reveal off-target interactions of a marketed kinase inhibitor in heart tissue. By enhancing scalability, reducing costs and enabling system-wide drug screening across a wide range of sample types, HT-PELSA—when combined with next-generation mass spectrometry—may offer a powerful platform poised to accelerate both drug discovery and basic biological research.

Protein function is often dynamically regulated by interactions with small molecules. Understanding the molecular mechanisms behind processes such as metabolic reactions and drug responses requires systematic techniques capable of probing protein–ligand interactions on a proteome-wide scale[1]. Among the various proteome-wide methods developed for mapping these interactions[2–5], PELSA[6] has recently emerged as a powerful tool owing to its simplicity and high performance. This approach detects twice as many interactions as other

techniques, while accurately pinpointing binding regions[6]. PELSA uses limited proteolysis[7] to identify protein regions stabilized by ligand binding. Unlike earlier mass spectrometry-based approaches[2,4], PELSA specifically analyzes peptides released from proteins after a brief trypsin digestion pulse, followed by the removal of undigested proteins using a molecular weight filter-based strategy.

The original PELSA workflow has two key limitations: it processes each sample individually, which restricts its throughput, and

[1]Molecular Systems Biology Unit, European Molecular Biology Laboratory (EMBL), Heidelberg, Germany. [2]Heidelberg University, Faculty of Medicine, and Heidelberg University Hospital, Institute for Computational Biomedicine, Heidelberg, Germany. [3]Proteomics Core Facility, European Molecular Biology Laboratory (EMBL), Heidelberg, Germany. [4]These authors contributed equally: Kejia Li, Clement M. Potel. ✉e-mail: mikhail.savitski@embl.de

it is performed on centrifuged lysates, confining the method primarily to cytoplasmic proteins. Here, we introduce an optimized, high-throughput adaptation, HT-PELSA, in which all steps are performed in 96-well plates at room temperature, improving throughput by 100-fold. This streamlined workflow enables the preparation of hundreds of samples per day, enhancing scalability and reproducibility while reducing costs without compromising performance. Moreover, we demonstrate that HT-PELSA expands PELSA's capabilities to measure membrane protein targets across a broader range of sample types, including cell lines, tissues and lower organisms. This advancement paves the way for more comprehensive profiling of protein–ligand interactions and offers powerful opportunities for both drug discovery and fundamental biological research.

## Results

### HT-PELSA workflow

Compared to the original PELSA workflow, HT-PELSA introduces several key improvements that enhance efficiency, scalability, reproducibility and cost effectiveness (Fig. 1a). (1) All steps are performed in 96-well plates instead of single tubes, substantially increasing throughput. (2) All 96 samples are processed simultaneously, unlike the original PELSA protocol, in which each sample was prepared sequentially owing to short incubation times. This reduces processing time up to 100-fold depending on the number of samples, enhances reproducibility, minimizes human error and ensures uniform ligand incubation time across all samples. (3) Samples are processed at room temperature, which eliminates the need for incubation at 37 °C, streamlining the procedure (Extended Data Fig. 1a and Supplementary Data 1). (4) The digestion time is extended to 4 min to ensure ease of operation without compromising performance (Fig. 1b, Extended Data Fig. 1a and Supplementary Data 1). (5) Intact, undigested proteins are removed using 96-well C18 plates, which selectively retain large protein fragments while allowing shorter peptides to elute. This replaces molecular weight cut-off filter single units, thus increasing throughput and eliminating the need for additional desalting before mass spectrometry analysis. Moreover, this procedure is compatible with crude lysates from cell lines, tissues and bacteria, which are samples that would otherwise clog filter membranes.

Overall, this workflow reduces sample processing time to under 2 h for up to 96 samples—with the possibility to process multiple plates in parallel—from cell lysis to mass spectrometry-ready peptides, offering a scalable, efficient and robust alternative to the original protocol. To validate the performance of HT-PELSA, we compared our high-throughput approach with the original method by identifying protein targets of staurosporine—a broad-spectrum kinase inhibitor commonly used as a benchmark for protein–ligand interaction studies—in K562 cell lysates. As shown in Fig. 1b, the total number of kinases identified as binding to staurosporine, as well as the method specificity, is comparable to that of the original protocol when analyzed

on the same instrument (Orbitrap Exploris), confirming the reliability and effectiveness of HT-PELSA. Notably, running the same samples on the new-generation mass spectrometer (Orbitrap Astral[8]) improves throughput threefold and increases the number of identified targets by 22% (Fig. 1b and Supplementary Data 1). A total of 93% of the significantly stabilized kinase peptides were localized within or near kinase domains (Extended Data Fig. 1b,c and Supplementary Data 1), thus confirming that HT-PELSA accurately maps ligand binding regions. Proteins are considered stabilized by the ligand if they become protected from digestion upon ligand binding. Accordingly, peptides that show decreased abundance in the treatment–control comparison are classified as stabilized peptides.

### Systematic determination of protein–ligand binding affinity

To understand the biological or therapeutic implications of a protein–ligand interaction, it is essential not only to identify the interaction but also to characterize its binding affinity. Our high-throughput approach facilitates the reproducible measurement of protein–ligand interactions across different ligand concentrations, allowing us to generate dose–response curves on a proteome-wide scale and determine half-maximum effective concentration ($EC_{50}$) values for each protein target. These insights into binding dynamics are crucial for drug development, facilitating the identification and optimization of potential therapeutic compounds.

To validate our methodology, we first determined the $-\log_{10}$-transformed $EC_{50}$ ($pEC_{50}$) values for kinase–staurosporine interactions in K562 cell lysates. HT-PELSA enables parallel processing of hundreds of samples, which results in highly reproducible $pEC_{50}$ values across replicates (median coefficient of variation, 2%), demonstrating its precision in quantifying ligand binding affinities (Fig. 1c,d, Extended Data Fig. 2a and Supplementary Data 2). Stabilized peptides—those less susceptible to tryptic digestion—showed high selectivity for kinases (90%; Fig. 1e). For the stabilized hit kinases, the majority of peptides exhibiting dose-dependent stabilization originate from the kinase domains, indicating that the dose–response data could accurately assign the binding region (Extended Data Fig. 2b,c). Among the non-kinase stabilized peptides, several known off-targets of staurosporine were identified, such as ferrochelatase (FECH)[3,9] and OSBPL3 (ref. 3) (Supplementary Data 2), underscoring how dose–response analysis increases confidence in target identification. Notably, destabilized peptides contained a lower proportion of kinase-derived sequences (Fig. 1f and Supplementary Data 2). However, those corresponding to kinases exhibited higher $pEC_{50}$ values compared to their non-kinase counterparts (Fig. 1f and Supplementary Data 2). This suggests that although stabilized peptides are generally more reliable for target discovery, incorporating dose–response curves enhances both the specificity and confidence of hits identified through destabilization. Moreover, it supports the observation seen in thermal proteome profiling (TPP)[3]

**Fig. 1 | HT-PELSA enables precise and accurate identification of drug targets and their binding affinities. a**, Comparison of the workflows between the HT-PELSA platform and the original PELSA. In the high-throughput platform, each step is performed in a 96-well format. **b**, The number of kinase and non-kinase targets identified in staurosporine-binding experiments in the original PELSA dataset[6] (Exploris), HT-PELSA (Exploris) and HT-PELSA (Astral) platforms are compared. All data were acquired in K562 cell lysates and processed uniformly using DIA-NN (v.1.8.1). The specificity of kinase targets among all identified targets is also indicated. **c**, The peptides of the Cyclin-G associated kinase (GAK) that exhibit a dose-dependent response to staurosporine ($R^2 > 0.9$, where $R^2$ denotes the correlation to a sigmoidal drug-response curve) are consistent across all four replicates. Error bars represent the standard deviation of fold change across the four PELSA replicates, and central dots indicate mean values ($n = 4$). These peptides, all derived from the kinase domain of GAK, show similar affinities for staurosporine, with $pEC_{50}$ values of 6.59, 6.89, 6.77 and 6.81. **d**, The coefficient of variation distribution of staurosporine $pEC_{50}$ values for all

stabilized peptides. Peptides were considered stabilized if they exhibited a dose-dependent response ($R^2 > 0.9$) and were at least 30% more resistant to trypsin at the highest staurosporine concentration across a minimum of three PELSA replicates. **e**, The staurosporine $pEC_{50}$ distribution of all stabilized peptides, categorized by whether they come from kinases or non-kinases. For all dose-responsive peptides, $pEC_{50}$ values represent the mean across replicates. **f**, As in **e**, but for all destabilized peptides. Destabilized peptides are defined as those with $R^2 > 0.9$ and at least 30% more susceptible to trypsin at the highest staurosporine concentration across a minimum of three replicates. **g**, Pearson correlation of staurosporine $pEC_{50}$ values obtained from HT-PELSA and kinobeads assay. Each dot represents a kinase identified in both datasets, with the number of kinases labeled. Kinases are grouped based on whether they are stabilized (green) or destabilized (red) by staurosporine in HT-PELSA. The dose–response staurosporine dataset was acquired using K562 cell lysates. Green shaded areas show the 95% confidence interval for the regression line (black).

that a small proportion of kinase targets are consistently destabilized upon inhibitor binding.

Leveraging the peptide-level resolution of our HT-PELSA, we found that some destabilized kinases are destabilized precisely at or near well-characterized protein-binding motifs. For example, BTK shows destabilization exactly at the SH3 and SH2 domains as well as around the CAV1-binding motif (Extended Data Fig. 2d). In MAP2K1, the kinase domain contains a unique RAF1-binding motif, and the peptide showing the strongest destabilization is located closest to this motif (Extended Data Fig. 2e). These findings suggest that a possible

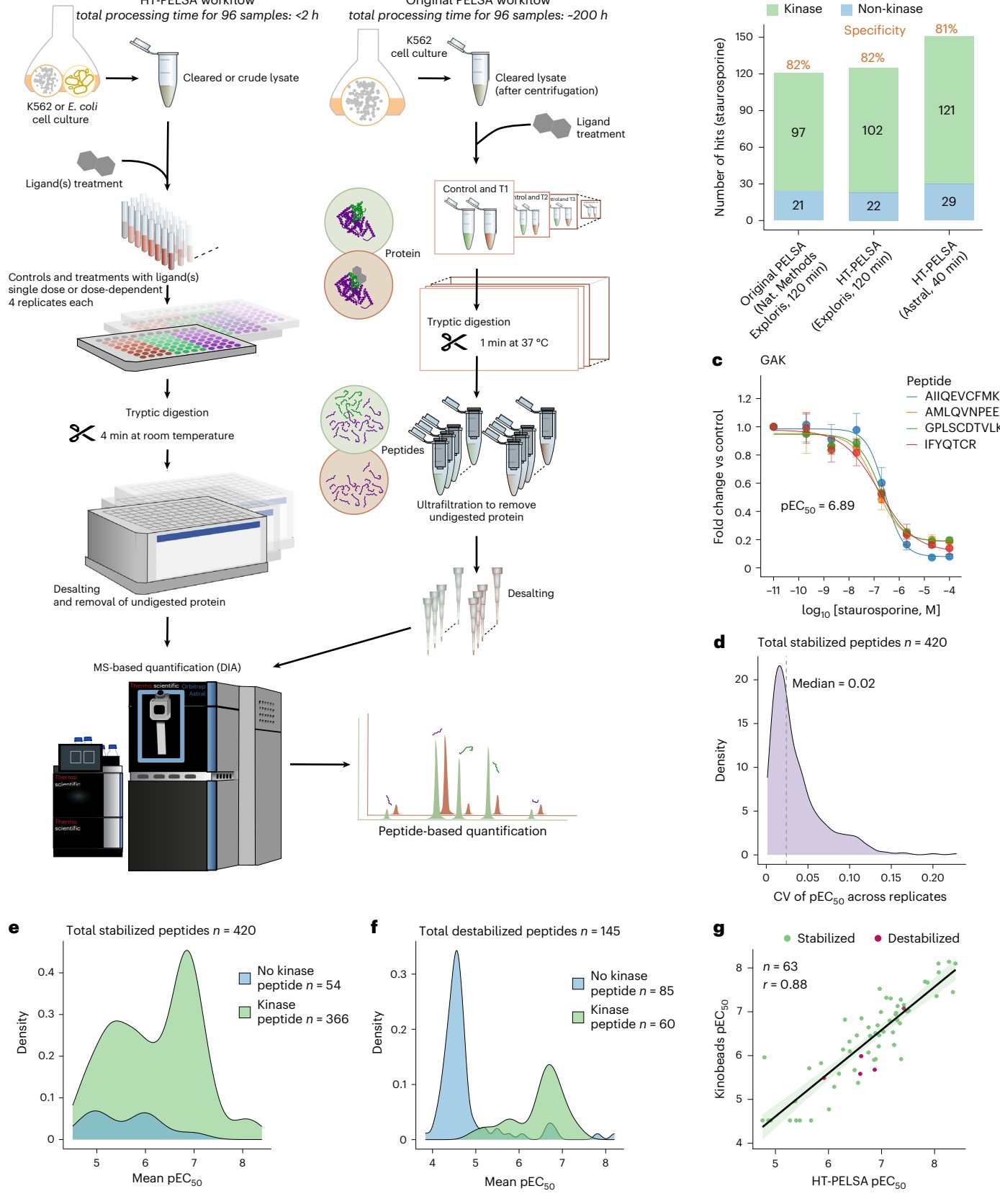

explanation for kinase destabilization could be disrupted protein–protein interactions.

Remarkably, the binding affinities of kinases measured by HT-PELSA closely aligned with values obtained from previously published kinobead competition assays[10,11], which is the gold standard for systematic measurement of kinase–inhibitor affinities. This agreement demonstrates that in addition to its high precision, HT-PELSA provides highly accurate affinity measurements for both stabilized and destabilized targets (Fig. 1g and Supplementary Data 2).

Next, we applied HT-PELSA to profile the binding affinity of the metabolite ATP. Using a dose–response assay, we identified a total of 1,426 stabilized peptides, enabling the characterization of ATP-binding affinities for 301 *Escherichia coli* proteins (Fig. 2a,b and Supplementary Data 3). A high proportion of stabilized peptides (1,013; 71%) and stabilized proteins (174; 58%) corresponded to UniProt-annotated ATP binders. The combination of HT-PELSA with the Orbitrap Astral system thus yields a resource that represents a substantial leap in coverage and specificity compared to the previous most comprehensive study systematically profiling protein–ATP interactions, which used limited proteolysis–mass spectrometry[12] (Fig. 2c, Extended Data Fig. 3a and Supplementary Data 3). This enhanced performance was also evident when taking into account a single high ATP concentration. At 5 mM ATP, HT-PELSA detected 172 known ATP-binding proteins with 61% specificity, whereas limited proteolysis–mass spectrometry detected 66 ATP binders (41% specificity) at the same concentration and 84 (36% specificity) at 25 mM ATP[12] (Extended Data Fig. 3b and Supplementary Data 3). In addition to treating lysates with ATP or vehicle and then aliquoting them into four replicates for separate HT-PELSA analyses, we also tested the performance of HT-PELSA when treating four separate lysate aliquots with ATP or vehicle. Our results showed that separate treatment replicates produce results that are as good as when treatment is performed before aliquoting (191 UniProt-annotated ATP binders and 58% specificity for dose–response result; 185 UniProt-annotated ATP binders and 61% specificity for single concentration 5 mM result) (Extended Data Fig. 3a,b and Supplementary Data 4).

The comprehensive, dose-dependent measurements with HT-PELSA also allow for the precise determination of $pEC_{50}$ values for the ATP-stabilized proteins identified (Fig. 2b, Extended Data Fig. 3c and Supplementary Data 3). Similar to the case of staurosporine and kinases, a smaller proportion of destabilized peptides corresponds to annotated ATP-binding proteins and exhibits, on average, higher $pEC_{50}$ values (Extended Data Fig. 3d,e and Supplementary Data 3). Analysis of AlphaFold[13] structures for proteins with known ATP-binding sites revealed that stabilized peptides are significantly closer to these binding sites than peptides with no detectable changes (Fig. 2d). Interestingly, this trend is much less pronounced—and not statistically significant—for destabilized peptides (Fig. 2d), suggesting that these could reveal allosteric effects induced by ATP binding in distal regions

of proteins. Gene Ontology analysis revealed that the destabilized proteins show significant enrichment for complex-related Cellular Component terms and chaperone protein unfolding-related Biological Process terms (Extended Data Fig. 4a,b). This suggests that PELSA can capture not only direct ligand targets but also changes in protein–protein interactions resulting from treatment. Supporting this observation, BioGRID data showed that destabilized proteins interact more frequently with each other or with stabilized proteins than pairs of unresponsive proteins (7.8% for destabilized protein pairs and 6.3% for destabilized–stabilized protein pairs compared to 2.7% of unresponsive protein pairs) (Extended Data Fig. 4c).

As an example, the protein-folding activity of the HSP70-like chaperone DnaK depends on its ATP-regulated interaction with substrates, mediated by allosteric communication between the nucleotide-binding domain and the substrate-binding domain, with substrate release triggered by ATP binding[14]. We observe that DnaK is stabilized with high affinity at the ATP-binding site and destabilized with lower affinity in the substrate-binding region (Fig. 2e,f and Supplementary Data 3). This observation supports the notion that substrate release is triggered by ATP binding at sufficiently high concentrations, making the non-bound substrate-binding domain more accessible to trypsin. Among the non-annotated ATP binders, we identify GuaC, which shows both stabilization and destabilization in two distinct regions at nearly identical ATP concentrations (Fig. 2g,h and Supplementary Data 3). In *Trypanosoma brucei*, GuaC has been shown to be inhibited by ATP through allosteric regulation[15], and similar inhibition of GuaC activity by ATP has been previously reported in *E. coli*[16]. Finally, we identify YjgR, an uncharacterized protein with a P-loop NTPase domain, which is strongly stabilized by ATP. Notably, the stabilized peptides are located within or adjacent to the P-loop region (Fig. 2i,j and Supplementary Data 3). In summary, our approach provides precise determination of ATP-binding affinities for both known and previously unannotated ATP-binding proteins. Furthermore, the high sequence-level resolution of our data not only detects binding events at ATP-binding sites but also offers valuable insights into allosteric regulation.

## Identification of membrane protein targets

Membrane proteins serve as the initial mediators of signaling cascades, making them prime targets for drug development. Originally, PELSA was performed on cleared lysates to avoid clogging of the molecular weight cut-off filter, meaning that these proteins were mostly removed by centrifugation before the ligand-binding assay. Here, we demonstrate that the plate-based HT-PELSA can be performed directly on crude lysates, preserving membrane proteins and expanding the assay's applicability (Fig. 3a).

As proof of concept, we first identified targets of dasatinib, a clinically approved tyrosine kinase inhibitor used to treat certain types of leukemia. Although its primary target, BCR-ABL1, is an intracellular

---

**Fig. 2 | HT-PELSA enables high-resolution characterization of ATP-binding proteins in *E. coli*. a**, Heatmap showing the $\log_2$(fold change) (FC) (relative to control) of 301 proteins exhibiting dose-dependent stabilization by ATP. Proteins are annotated based on whether they are UniProt-annotated ATP-binding proteins and are ranked by their apparent ATP-binding affinity ($pEC_{50}$). **b**, Distribution of $pEC_{50}$ values for all peptides stabilized by ATP (defined as $R^2 > 0.9$ and ≥30% increased resistance to trypsin at the highest ATP concentration in at least three replicates), grouped by whether they originate from UniProt-annotated ATP-binding proteins. **c**, Comparison of the number of peptides exhibiting dose-dependent responses to ATP in a published limited proteolysis–mass spectrometry (LiP–MS) dataset[12] and HT-PELSA, grouped by whether they are derived from UniProt-annotated ATP-binding proteins. **d**, Distributions of the Euclidean distance between peptide cleavage sites and UniProt-annotated ATP-binding residues for peptides belonging to known ATP binders, grouped by stabilized peptides ($n = 799$), unchanged peptides ($n = 4,938$) and destabilized peptides ($n = 81$). Statistical significance was assessed using a two-sided Wilcoxon rank-sum test (no adjustment). Box plots

show the median (line), interquartile range (IQR, box) and ±1.5× IQR (whiskers); outliers are omitted for clarity. **e**, Structural representation of DnaK (PDB 4B9Q), with stabilized peptides highlighted in blue and destabilized peptides in orange. ATP is shown as yellow spheres. **f**, $pEC_{50}$ values of peptides from DnaK stabilized and destabilized by ATP across HT-PELSA replicates ($n = 4$ for peptides 503–514, 422–445 and 346–352; peptide 352–362 was quantified only in three replicates, thus $n = 3$); each dot represents one HT-PELSA replicate (HT-PELSA replicates refer to treating lysates with compound or vehicle and then aliquoting into replicates for separate HT-PELSA analyses). Error bars indicate standard deviations, and central lines indicate mean values. **g**, Structural model of GuaC (AlphaFold: AF-P60560), with nine peptides stabilized by ATP shown in blue and two destabilized peptides in orange. **h**, Summary of mean $pEC_{50}$ values and coefficients of variation (CV) across at least three HT-PELSA replicates for peptides displayed in **g**. **i**, Structural model of YjgR (AlphaFold: AF-P39342), with the P-loop NTPase domain colored in red and two peptides stabilized by ATP shown in blue. **j**, Summary of mean $pEC_{50}$ values and coefficients of variation across at least three HT-PELSA replicates for peptides shown in **i**.

kinase, many of dasatinib's off-targets are membrane-associated proteins[17]. HT-PELSA performed on crude K562 cell lysates reveals 556 additional proteins compared to cleared lysates (Fig. 3b,c and Supplementary Data 5). These proteins are significantly enriched for

Gene Ontology terms related to mitochondrial and plasma membrane functions (Extended Data Fig. 5a). Our results show that the primary target, BCR-ABL1, stands out as the top target for both cleared and crude lysates (Fig. 3b,c and Supplementary Data 5). In addition to

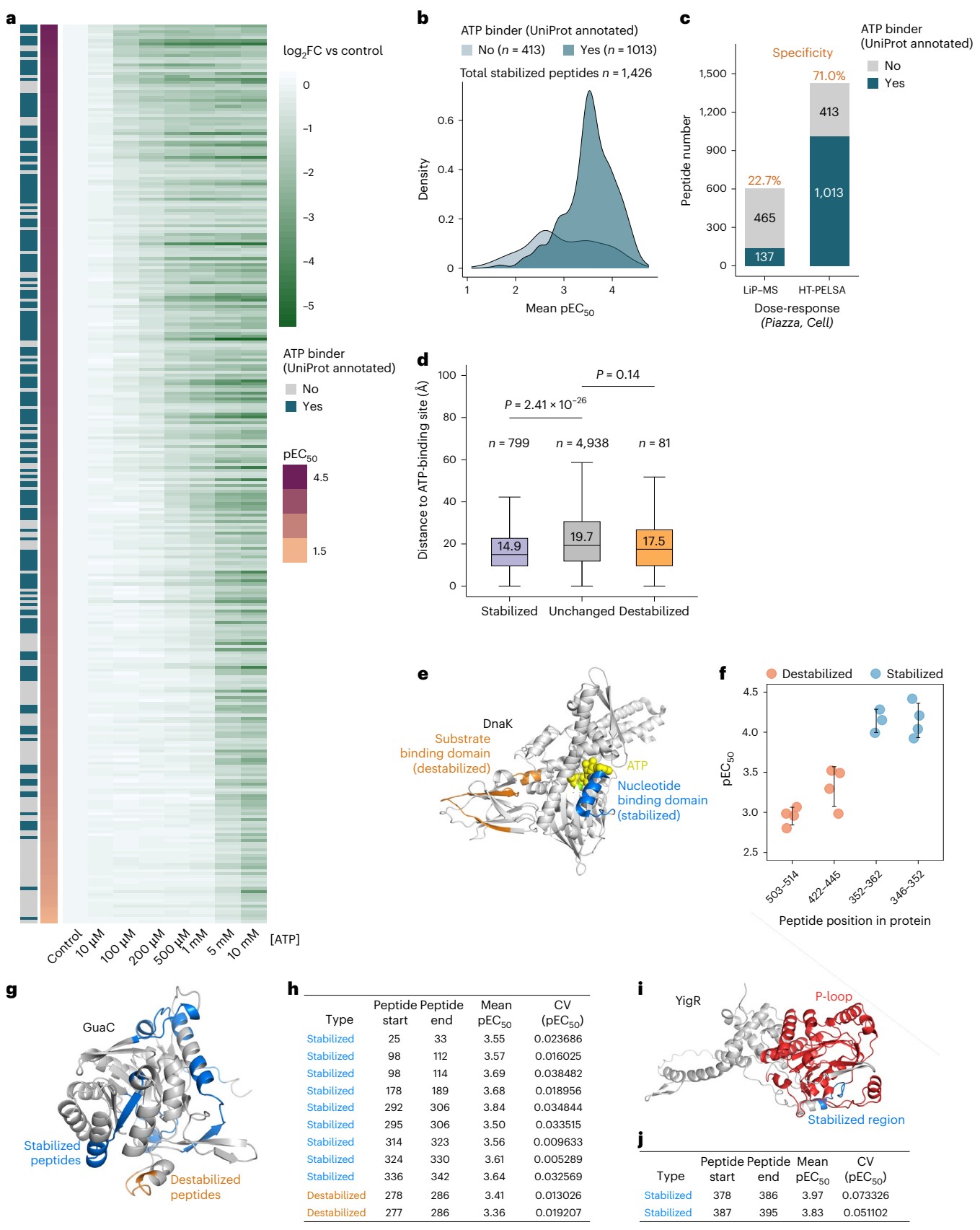

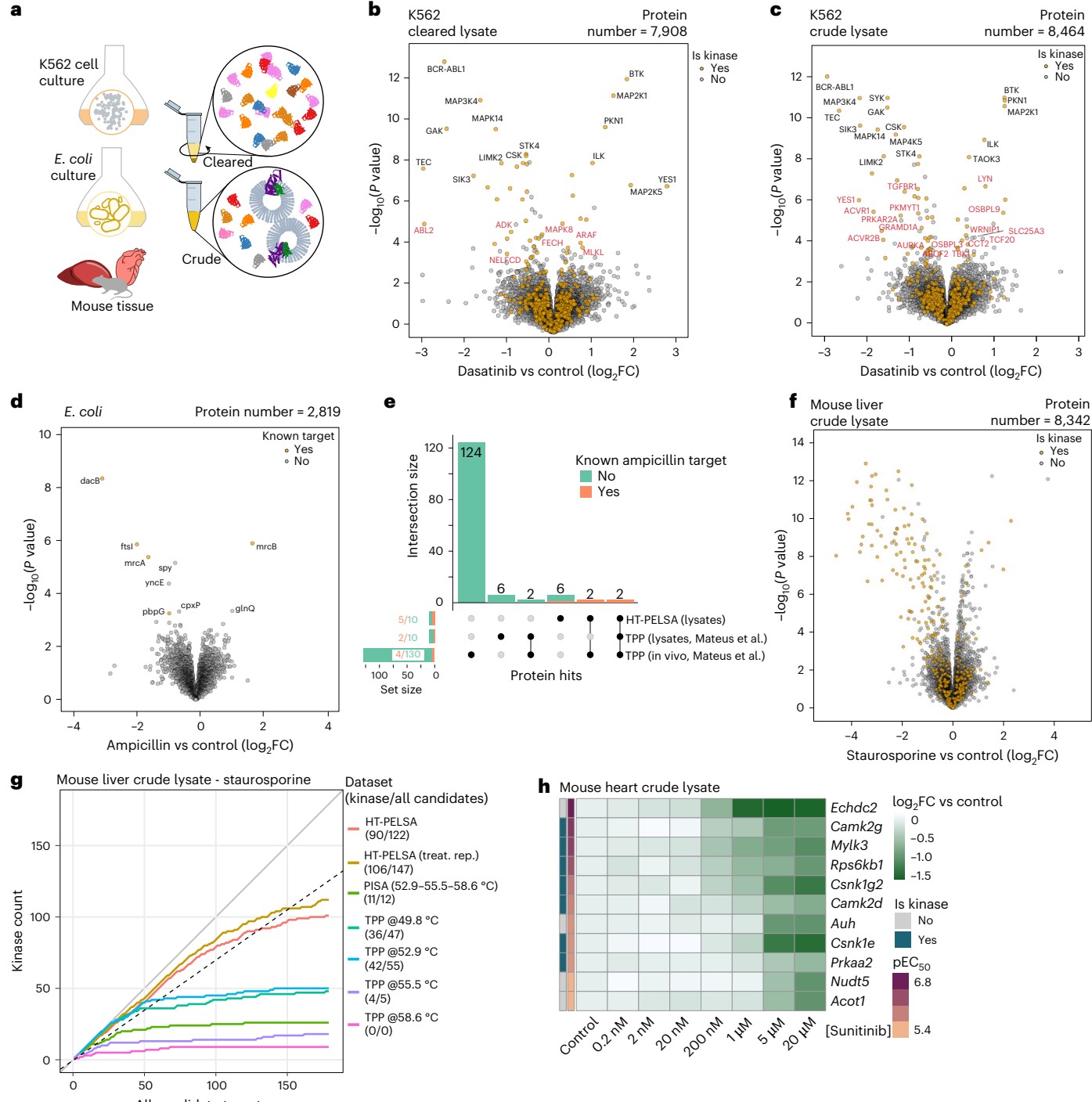

**Fig. 3 | HT-PELSA extends target identification to membrane proteins in cell lines, mouse tissues and bacteria. a**, Schematic representation of crude or cleared K562 cells, *E. coli* and tissue lysates for HT-PELSA analysis. **b**, Volcano plot showing candidate dasatinib-binding targets identified by HT-PELSA in cleared K562 lysates treated with 5 µM dasatinib. Proteins were considered candidate targets if they met the threshold of $-\log_{10}(P) > 4$. For each protein, the peptide with the lowest *P* value is displayed. **c**, As in **b**, but for crude K562 lysates. Protein targets that are uniquely identified in the cleared lysates or crude lysates are annotated with gene names in red. **d**, Volcano plot showing candidate ampicillin-binding targets identified by HT-PELSA in crude *E. coli* lysates exposed to 10 µM ampicillin. Proteins with $-\log_{10}(P) > 3.4$ are annotated with gene names. **e**, UpSet plot showing overlap of hits following 10 µM ampicillin treatment in *E. coli* lysate by HT-PELSA and *E. coli* lysate and whole cells by TPP experiments. **f**, Volcano

plot showing candidate staurosporine-binding targets identified by HT-PELSA in crude mouse liver lysates exposed to 20 µM staurosporine. **g**, True positive rate evaluation for HT-PELSA (with treatment performed before aliquoting lysate or treatment performed separately for each replicate), TPP and PISA experiments (both with treatment performed separately for each replicate). The gray line (slope = 1) and black dashed line (slope = 0.7) represent scenarios in which 100% and 70% of the candidate targets are kinase targets, respectively. **h**, Heatmap displaying $\log_2$(fold change) (versus control) of 11 proteins that show dose-dependent stabilization to sunitinib in mouse heart lysate. The gene names of the proteins are labeled. In **b**, **c**, **d** and **f**, *P* values were determined by a two-sided empirical Bayes *t*-test without adjustment (four HT-PELSA replicates; HT-PELSA replicates refer to treating lysates with compound or vehicle and then aliquoting into replicates for separate HT-PELSA analyses).

BCR-ABL1, we identified 43 shared targets in both crude and cleared lysates, 36 of which are protein kinases, highlighting the promiscuity of dasatinib (Extended Data Fig. 5b). However, among these 43 targets, only three are transmembrane proteins (Extended Data Fig. 5c). Performing HT-PELSA on crude lysates resulted in an increase in identification of membrane-associated target proteins, including several known dasatinib off-targets such as LYN, TGFBR1, YES1, ACVR1 and ACVR2B[9] (Fig. 3c, Extended Data Fig. 5b,c and Supplementary Data 5). Notably, the kinase YES1 was both stabilized at the kinase domain and destabilized at the SH2 domain (the protein–protein interacting domain) (Extended Data Fig. 5d and Supplementary Data 5). Consistent with the staurosporine experiment, we also see destabilization of BTK and MAP2K1, with destabilized peptides occurring within or proximate to the annotated protein interaction domains (Extended Data Fig. 5e,f and Supplementary Data 5).

To further emphasize the ability to characterize membrane protein interactions, we next sought to identify ampicillin targets in *E. coli* crude lysate. Ampicillin is a penicillin derivative antibiotic that binds to inner membrane penicillin-binding proteins, which are essential for cell membrane biosynthesis. HT-PELSA identified five known targets of ampicillin (*mrcA*, *mrcB*, *ftsI*, *dacB* and *pbpG*; Fig. 3d and Supplementary Data 6). By comparison, TPP in *E. coli* lysate identified two known targets, while TPP in live cells identified four known targets[18]. Notably, all known targets identified by TPP—both in lysates and in vivo—are included among the five known targets identified by HT-PELSA; the additional targets identified in vivo by TPP (Fig. 3e) should be a result of downstream effects of inhibition of the main targets. The identified known ampicillin-binding proteins belong to three protein families: penicillin-binding proteins, D-alanyl-D-alanine carboxypeptidases and peptidoglycan D,D-transpeptidases. Notably, the penicillin-binding proteins encoded by *mrcA* and *mrcB* contain an amino-terminal transglycosylase domain, annotated in UniProt as penicillin-insensitive, and a carboxy-terminal transpeptidase domain, annotated as penicillin-sensitive. Consistently, our HT-PELSA results show that the proteins encoded by *mrcA* and *mrcB* are specifically perturbed at their C-terminal transpeptidase domains instead of their N-terminal domains (Extended Data Fig. 5g,h and Supplementary Data 6). We can conclude that HT-PELSA can very sensitively identify targets and binding regions in lysates, while TPP can confirm that these interactions happen inside cells and additionally reveal downstream effects. These findings also demonstrate that the HT-PELSA protocol effectively captures protein–ligand interactions in crude lysates of bacterial samples, expanding its applicability to microbial systems.

### HT-PELSA in mouse tissue reveals kinase inhibitors off-targets

Mapping protein–ligand interactions in tissue provides more biologically relevant insights than in cell lines. However, studying these interactions is more challenging because of the heightened complexity and potential biomolecular contaminants, which can affect both the number and specificity of target identifications. For example, a recent study detected only a handful of kinases binding to staurosporine in rat liver tissue using proteome integral solubility alteration (PISA)[19]. By contrast, HT-PELSA successfully identified 90 kinases in a mouse liver crude lysate, demonstrating comparable performance in both cell lines and tissues (Fig. 3f,g and Supplementary Data 7). A total of 106 kinase targets were obtained when treating the crude lysate aliquots separately with staurosporine and performing HT-PELSA (Fig. 3g, Extended Data Fig. 6a and Supplementary Data 8), again showing that whether treatment is performed together or separately does not significantly impact the performance of HT-PELSA.

To enable a parallel comparison of HT-PELSA with existing methods for target identification in crude tissue lysates, we also performed PISA and TPP to identify staurosporine targets in mouse liver lysates, using the same experimental settings as in HT-PELSA (treatments performed separately for different replicates). PISA samples were obtained by pooling the supernatants after heat treatment at 52.9 °C, 55.5 °C and 58.6 °C, as suggested by the PISA tissue study[19]. In total, 4,244 proteins were quantified for the PISA. As a reference, ~3,500 proteins were quantified for rat liver in the previously reported PISA dataset[19], confirming the adequate proteome coverage of our PISA dataset. The smaller number of proteins detected in PISA compared to HT-PELSA (~8,400 proteins) is probably because of the greater complexity of tissue lysates: abundant extracellular-matrix and other connective-tissue components can keep proteins insoluble and make them easy to precipitate (an essential step in TPP). The volcano plots and specificity curves (Fig. 3g and Extended Data Fig. 6b) clearly show that although the top hits in PISA are kinases, the number of kinases identified at 70% specificity is significantly higher for HT-PELSA (106 vs 11 kinase hits) (Supplementary Data 8).

Analysis of individual temperatures revealed that the performance of TPP is best at 52.9 °C (Fig. 3g, Extended Data Fig. 6c–f and Supplementary Data 8), showing the best specificity and highest kinase coverage (42 kinase hits at 70% specificity), in line with previous work[3,20], followed by 49.8 °C (36 kinase hits at 70% specificity), a temperature at which the proteome and kinase coverage is higher but the kinase fold changes are much smaller, given that most of them have not melted at that temperature (Fig. 3g, Extended Data Fig. 6c and Supplementary Data 8). The worst results in terms of proteome coverage and kinase hit identification were obtained at 55.5 °C and 58.6 °C (four and zero kinases at 70% specificity) (Fig. 3g, Extended Data Fig. 6e,f and Supplementary Data 8). Taken together, this shows that HT-PELSA is a very sensitive method for identifying drug targets in the complex tissue lysates. TPP can identify drug targets with good specificity in tissue, but twofold less than HT-PELSA. It also highlights that care needs to be taken when performing experiments in the PISA format, given that including temperatures at which the proteome coverage is low and the response to treatment is minimal in the pool will adversely affect the results, even when a few temperatures are pooled.

Identifying drug targets and their affinities, in addition to the primary target, is crucial in drug discovery, as off-target interactions can result in harmful side effects. To illustrate this point, we applied HT-PELSA to characterize targets of sunitinib, a clinically approved multi-targeted tyrosine kinase inhibitor used to treat renal, gastrointestinal and pancreatic cancers. Clinical studies have reported direct cardiomyocyte toxicity associated with sunitinib treatment, which can progress to heart failure in affected patients[21]. In this study, we identified four known kinase off-targets of sunitinib in mouse heart tissue: the α subunit Prkaa2 of Ampk, Camk2g, Csn1ke and Csnk1g2 (ref. 17) (Fig. 3h and Supplementary Data 9). Inhibition of Ampk in the heart has been recognized as a key driver of sunitinib-induced cardiotoxicity by reducing mitochondrial ATP production[22]. We also characterized additional direct high-affinity off-targets of sunitinib (Fig. 3h). These include Camk2d, a kinase with heart-specific expression[23] that has so far only been reported to bind sunitinib in a recombinant kinase assay[24] and is involved in the regulation of heart function[25], as well as myosin light chain kinase 3 (Mylk3), which is known to have cardiomyocyte-specific expression[26], underscoring the importance of tissue-specific drug target profiling. Mylk3 activity is essential for maintaining basal contractile function in the heart[27,28], potentially revealing additional mechanisms underlying sunitinib-induced cardiotoxicity. In addition, several mitochondrial proteins (Echdc2, Auh, Acot1) were identified as direct high-affinity targets of sunitinib (Fig. 3h). Given that mitochondrial functions are essential for energy homeostasis, their disruption may contribute to mitochondrial dysfunction and, ultimately, cardiac pathologies.

## Discussion

The recently introduced PELSA workflow enables sensitive and systematic probing of protein–ligand interactions. In this study, we develop HT-PELSA, a workflow that dramatically increases sample throughput

and streamlines sample preparation. These advancements pave the way for high-throughput screening of protein–small molecule interactions, providing information on protein targets, binding affinities and protein binding regions. Furthermore, the introduced HT-PELSA protocol extends protein–ligand profiling beyond the cleared cell line lysate setting. We demonstrated that HT-PELSA accommodates a broad range of sample types, including crude tissue and bacterial lysates, and enables mapping of ligand interactions with membrane proteins, a key class of proteins in drug discovery.

HT-PELSA is an in-lysate method; therefore, combining it with methods capable of systematic protein–ligand mapping in intact cells, such as TPP, will be highly impactful for drug discovery. The sensitive ligand-binding region detection enabled by HT-PELSA—when combined with in situ information on ligand binding from TPP, as well as the downstream effects that TPP can identify—will be very valuable for characterizing the mechanisms of action of small molecules.

We expect that the versatility with respect to source material and sensitivity of ligand binding detection of HT-PELSA will make it the method of choice for finding targets of uncharacterized compounds in lysate. However, if the goal is to assess compound selectivity within a specific enzyme family, such as kinases, a kinase-specific method like kinobeads will still offer higher sensitivity but will generally not be able to detect non-kinase off-targets of kinase-specific inhibitors.

It is noteworthy that some direct targets are destabilized in their ligand-binding domains in HT-PELSA experiments. This could potentially be explained by ligand binding competing away interacting proteins, which could lead to destabilization if interactions were taking place at or close to the binding domain and occluding it. A mechanistic investigation of such cases will be a subject of future studies.

HT-PELSA unlocks exciting possibilities. The 96-well plate format, extended digestion time of 4 min and removal of the short 37 °C incubation make the workflow primed for future automation. With higher sensitivity than alternative approaches—currently requiring 60 μg of protein input per sample, with only 4% of the resulting peptides being injected for mass spectrometry analysis—the input could be further scaled down by decreasing the starting protein concentration or volume and optimizing digestion conditions accordingly. This makes HT-PELSA particularly advantageous for studying protein interactions in limited or valuable materials, such as clinical samples. Finally, in the future, the scope of HT-PELSA could be expanded to systematically investigate protein–protein or protein–nucleic acid interactions, further broadening its impact in biomolecular research.

## Online content

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

## Methods

All animal care and procedures were conducted in line with EMBL regulations and guidelines for the use of animals in research and were reviewed and approved by the Institutional Animal Care and Use Committee (IACUC). All mouse experiments were performed using approved protocols by the EMBL Ethics Committee (license 21-002_HD_MZ).

### Human cell lines

All cell lines used in this study were verified to be negative for *Mycoplasma* contamination. K562 cells (obtained from the American Type Culture Collection, CCL-243) were cultured in RPMI 1640 medium (Thermo Fisher) supplemented with 2 mM L-glutamine and 10% FBS at 37 °C (5% $CO_2$). For collection, K562 cells were pelleted at 1,000$g$ for 5 min. The collected cells were washed three times with ice-cold PBS and stored at −80 °C for further analysis.

### *E. coli*

*E. coli* K-12 strain BW25113 cells were grown overnight at 37 °C in lysogeny broth (Lennox) and diluted 100-fold into 20 ml of fresh lysogeny broth. Cultures were grown aerobically at 37 °C with shaking until reaching an optical density at 578 nm of 2. For collection, *E. coli* were pelleted at 1,000$g$ for 5 min, washed three times with ice-cold PBS and stored at −80 °C for further analysis.

### Mouse tissues

For the mouse liver collection, germ-free C57BL/6 mice were maintained and bred in air-conditioned gnotobiotic isolators (CBC) (temperature, 22 ± 2 °C; relative humidity, 50 ± 10%) with a 12 h light–dark cycle. Mice were provided with standard, autoclaved chow (1318P FORTI, Altromin) ad libitum. After reaching maturity at 7 weeks, a male mouse was colonized with laboratory strains from a cryopreserved inoculum composed of *Akkermansia muciniphila* DSM 22959, *Bacteroides uniformis* DSM 6597, *Clostridium sporogenes* ATCC 15579, *Eubacterium rectale* DSM 17629, *Lactobacillus gasseri* DSM 20243, *Parabacteroides distasonis* DSM 20701, *Segatella copri* DSM 18205 and *Ruminococcus gnavus* ATCC 29149. The colonized animal was single-housed for 14 days before it was killed by carbon dioxide exposure, and liver samples were collected and stored at −70 °C until further analysis.

For the mouse heart collection, C57BL/6J mice were maintained in individually ventilated, air-conditioned plastic cages (Tecniplast) (temperature, 22 ± 2 °C; relative humidity, 50 ± 10%) under a 12 h light–dark cycle, with ad libitum access to 1318P autoclavable diet (Altromin). After 4 months, a male mouse was killed by carbon dioxide exposure, and heart samples were collected and stored at −70 °C until further analysis.

### Lysate preparation

K562 cells were resuspended in ice-cold lysis buffer (PBS supplemented with 1% (v/v) protease inhibitor cocktail; Sigma-Aldrich, P8340-5ML) and lysed by three cycles of snap-freezing in liquid nitrogen followed by thawing in a 37 °C water bath. To obtain cleared lysates, cell extracts were centrifuged at 20,000$g$ for 10 min at 4 °C to remove cellular debris. For crude lysates, non-centrifuged cell extracts were used directly for subsequent analysis. Using multiple freeze–thaw cycles for cell lysis in PBS at a low lysate concentration (1.2 mg ml$^{-1}$ of protein) without detergent ensures that the released DNA is sufficiently sheared and present at low concentration. This prevents the formation of the sticky, viscous texture typically caused by long strands of intact or aggregated DNA released under harsher lysis conditions.

*E. coli* cells were resuspended in lysis buffer containing PBS, 1% (v/v) protease inhibitor cocktail, 50 μg ml$^{-1}$ lysozyme (Sigma-Aldrich, L7386), 0.25% (v/v) benzonase (Merck, 71206-3) and 2 mM MgCl$_2$. The suspension was incubated at 21 °C for 20 min to facilitate cell wall degradation. Following incubation, cells were further lysed by three cycles of freeze–thaw. Although the crude *E. coli* lysates contained insoluble debris, they were homogeneous and could be evenly distributed for

different treatments and replicates. For ampicillin HT-PELSA analysis, the resulting crude *E. coli* lysates were used directly. For ATP HT-PELSA analysis, the lysis buffer was supplemented with an additional 10 mM MgCl$_2$ to promote ATP–Mg$^{2+}$ complex formation at the maximum working ATP concentration of 10 mM. To enable accurate determination of the binding affinity to ATP, lysates were desalted using size-exclusion chromatography (SEC; Zeba Spin Desalting Columns 7 kDa molecular weight cut-off; Thermo Fisher Scientific) to remove endogenous ATP. Before desalting, the crude *E. coli* lysates were spun down at 500$g$, 4 °C for 10 min to remove intact cells that could potentially block the SEC columns; the cell debris can pass through the SEC desalting column and stay in the desalted lysates. The resulting desalted *E. coli* lysates were used for subsequent analysis.

The mouse liver tissue was resuspended in lysis buffer containing PBS and 1% (v/v) protease inhibitor cocktail, then homogenized using a bead beater (Beadruptor Elite, Omni International) with a mix of 2.8 mm and 1.6 mm zirconium silicate beads for three cycles of 20 s at 4 m s$^{-1}$, with 30 s cooling between cycles. The resulting lysates were used for subsequent analysis. Mouse heart lysates were prepared similarly to liver lysates, but with five cycles of bead beating for 20 s at 6 m s$^{-1}$, with 30 s cooling intervals between cycles. Both mouse tissue lysates can be evenly distributed.

For all samples, the protein concentrations of lysates were measured with a Rapid Gold BCA Protein Assay Kit (Thermo Scientific) and adjusted to 1.2 mg ml$^{-1}$ with the lysis buffer.

### Incubation with the ligands

In benchmarking experiments against the original PELSA protocol[6], a fixed staurosporine concentration of 20 μM was used for both HT-PELSA protocols (4 min and 3 min digestion at 22 °C). For the staurosporine dose–response assay, concentrations tested were 0.2 nM, 2 nM, 20 nM, 200 nM, 2 μM, 20 μM and 100 μM. In the *E. coli* ATP dose–response experiment, ATP concentrations of 10 μM, 100 μM, 200 μM, 500 μM, 1 mM, 5 mM and 10 mM were used. In the dasatinib experiment, both crude and cleared K562 cell lysates were incubated with 5 μM dasatinib. In the *E. coli* ampicillin-binding assay, crude *E. coli* lysates were incubated with 10 μM ampicillin. In the mouse liver staurosporine-binding experiment, lysates were incubated with 20 μM staurosporine. In the mouse heart sunitinib dose–response experiment, concentrations of 0.2 nM, 2 nM, 20 nM, 200 nM, 1 μM, 5 μM and 20 μM were used. Stock solutions (100-fold) of staurosporine, dasatinib and sunitinib were prepared in dimethyl sulfoxide (DMSO), with equivalent DMSO volumes added to the control group. Stock solutions (50-fold) of ATP and ampicillin were prepared in water, with equivalent water volumes added to the control group. The pH of the ATP stock solution was adjusted to 7 with sodium hydroxide before use. The lysates were incubated with the different ligands at 25 °C for 30 min.

To test whether aliquoting replicates before or after compound treatment affects HT-PELSA performance, we also performed *E. coli* ATP experiments and staurosporine liver experiments with replicates aliquoted before treatment. Specifically, 50 μl of lysate was incubated in parallel with the above ATP (or staurosporine) concentrations or vehicle control for 30 min at 25 °C in a 96-well plate, using four replicates.

### HT-PELSA workflow

After incubation, each sample was divided into four 50 μl replicates in a 96-well plate. Using a Gilson Platemaster, all samples were transferred to a second 96-well plate pre-loaded with 5 μl of trypsin (Sigma-Aldrich, T1426) stock solution (5 mg ml$^{-1}$). The trypsin was kept on ice and added to the plate shortly before sample addition. The samples were mixed with trypsin by pipetting up and down for 30 s and incubated at 22 °C for an additional 3 min and 30 s. Digestion was quenched by adding a threefold volume (165 μl) of 8 M guanidine hydrochloride (GdmCl) solution (containing 8 M GdmCl and 60 mM HEPES, pH 8.2), resulting in a final GdmCl concentration of 6 M. Control samples were

subsequently supplemented with the same concentration of ligand as their corresponding ligand-treated samples to account for potential interference with ionization during mass spectrometry analysis. Next, 10 mM of Tris(2-carboxyethyl)phosphine and 40 mM of chloroacetamide (final concentrations) were added to the samples, and the mixture was then heated at 95 °C for 5 min for cysteine carbamidomethylation. Alternatively, reduction–alkylation can be performed at a lower temperature and a longer incubation time.

After carbamidomethylation, the samples were acidified with trifluoroacetic acid (TFA) to a final concentration of 1%. Before loading onto the 100 mg Sep-Pak tC18 96-well plate, the plate was conditioned as follows, with every step performed at room temperature. First, it was centrifuged at 3,000g for 5 min without liquid to account for potential differences in C18 packing that could affect flow rate. Each well was then washed twice with 1 ml of 100% ACN (centrifuged at 20g for 1 min), followed by two washes with 1 ml of 0.1% TFA (centrifuged at 100g for 1 min with an acceleration setting of five out of nine). Samples were then loaded onto the plate (centrifuged at 100g for 1 min with acceleration set to five out of nine) and washed twice with 1 ml of 0.1% TFA (centrifuged as before). Finally, peptides were eluted with 2 × 100 µl of 50% ACN, 0.1% TFA (centrifuged at 20g for 1 min), followed by a final centrifugation step at 500g for 1 min. The eluted peptides were collected directly onto a mass spectrometry-compatible 96-well plate and dried. To ensure that the tips of the C18 plate do not come into contact with the bottom of the well, a 96-tip holder from a pipette tip box was used between the C18 plate and the collection plate. Alternatively, only a fraction of the elution can be dried, as only 4% of the eluate was injected into the liquid chromatography–tandem mass spectrometry (LC–MS/MS) system.

## TPP and PISA with staurosporine

Mouse liver lysates (prepared as described above) were diluted to a protein concentration of 3.5 mg ml$^{-1}$. Four aliquots were incubated with 20 µM staurosporine or DMSO as a control for 30 min at 25 °C. For each treatment replicate, 20 µl aliquots were distributed on a 96-well PCR plate, and the aliquots were treated with a gradient at the different temperatures (49.8 °C, 52.9 °C, 55.5 °C, 58.6 °C) for 3 min. After a 3 min incubation at room temperature, samples were put on ice, and 30 µl lysis buffer, including final concentrations of 0.8% NP-40, 1.5 mM MgCl$_2$, 0.25 U benzonase and 1× protease cocktail, was added to each sample and incubated for 1 h in the cold room. To remove aggregates, samples were centrifuged on a 0.45 µm filter plate (Millipore, MSHVN4550) and the soluble fraction was collected. The protein concentration at the lowest temperature was determined, and a volume equal to 5 µg was used for overnight protein digestion with trypsin and Lys-C using a modified SP3 protocol[18,29]. For PISA, aliquots of three temperature samples (52.9 °C, 55.5 °C and 58.6 °C) were pooled. For the TPP experiments, the four temperatures were kept separate.

## LC–MS/MS analysis

The samples were resuspended in a loading buffer containing 0.1% TFA and 4% acetonitrile in MS-grade water; a volume corresponding to 5% of the sample was injected. For both LC systems, solvent A comprised 0.1% formic acid supplemented with 3% DMSO in LC–MS-grade water, and solvent B comprised 0.1% formic acid supplemented with 3% DMSO in LC–MS-grade acetonitrile. For the HT-PELSA Exploris 480 analysis, peptides were separated using an UltiMate 3000 RSLCnano system (Thermo Fisher Scientific) equipped with a trapping cartridge (Precolumn; C18 PepMap 100, 5 µm, 300 µm inner diameter × 5 mm, 100 Å) and an analytical column (Waters nanoEase HSS C18 T3, 75 µm × 25 cm, 1.8 µm, 100 Å). Peptides were loaded onto the trapping cartridge (30 µl min$^{-1}$ solvent A for 3 min) and eluted with a constant flow of 300 nl min$^{-1}$. Peptides were separated using a linear gradient of 8–25% B for 99 min, followed by an increase to 40% B within 5 min before washing at 85% B for 4 min and re-equilibration to initial conditions.

The LC system was coupled to an Exploris 480 mass spectrometer (Thermo Fisher Scientific) operated in data-independent acquisition (DIA) mode. The instrument was operated in positive ion mode with a spray voltage of 2.5 kV and a capillary temperature of 275 °C. Full-scan MS spectra were acquired with a resolution of 60,000 with a mass range of 420–680 and an AGC target of $3 × 10^6$. DIA spectra were acquired in the Orbitrap mass analyzer with 4 $m/z$ windows between 430 and 670 $m/z$. The MS2 scan range was set to 200–1,800 $m/z$, the normalized collision energy was set to 28 and the default charge state was set at 2+. The normalized AGC target was set to 3,000% and the maximum injection time to auto.

For HT-PELSA Astral, TPP and PISA sample analysis, peptides were separated using a Vanquish Neo UHPLC system (Thermo Fisher Scientific) operated in trap-and-elute mode. The LC was equipped with a trapping cartridge (Precolumn; PepMap Neo C18, 5 µm, 300 µm inner diameter × 5 mm, 100 Å) and an analytical column (Ionopticks AUR3-25075C18-XT, 25 cm × 75 µm inner diameter, 1.7 µm C18). Solvent A comprised 0.1% formic acid supplemented with 3% DMSO in LC–MS-grade water, and solvent B comprised 0.1% formic acid supplemented with 3% DMSO in LC–MS-grade acetonitrile. Peptides were loaded onto the trapping cartridge and separated on the analytical column using a linear gradient of 5–26% B (flow rate of 300 nl min$^{-1}$) for 29.7 min, followed by an increase to 40% B within 3 min before washing at 85% B for 5 min and re-equilibration to initial conditions (total MS acquisition time of 40 min). The LC system was coupled to an Orbitrap Astral mass spectrometer (Thermo Fisher Scientific) operated in DIA mode. The instrument was operated in positive ion mode with a spray voltage of 1.8 kV and a capillary temperature of 280 °C. Full-scan MS spectra were acquired in the Orbitrap at a resolution of 240,000 with a mass range of 430–680 $m/z$, an AGC target of $5 × 10^6$ charges and a maximum injection time of 5 ms. DIA spectra were acquired in the Astral mass analyzer with 2 $m/z$ windows between 430 and 680 $m/z$. The MS2 scan range was set to 150–2,000 $m/z$, the normalized collision energy was set to 25 and the default charge state was set at 2+. The normalized AGC target was set to 500% and the maximum injection time was set at 5 ms.

## Data analysis

Raw files were analyzed using DIA-NN (v.1.8.1)[30] with the directDIA module using an in silico DIA-NN predicted spectral library (carbamidomethylation of cysteine, N-terminal M excision; two missed cleavages; precursor $m/z$ range, 430–680). The spectral library was generated from fasta files downloaded from UniProt (human: UniProt 2022 release, 20,311 sequences, reviewed; mouse: UniProt 2023 release, 17,173 sequences, reviewed; E. coli: UniProt 2024 release, 4,531 sequences, reviewed). The DIA-NN search used the following parameters: precursor FDR (%) = 1; mass accuracy, MS1 accuracy; scan window = 0; use isotopologues; MBR enabled; heuristic protein inference enabled; no shared spectra; protein inference = genes, neural network classifier, single-pass mode; quantification strategy, robust LC (high precision); cross-run normalization, RT-dependent; library generation, smart profiling; speed and RAM usage, optimal results.

For HT-PELSA analysis, the precursor matrix outputs from DIA-NN (v.1.8.1) were used for downstream statistical analysis in R. Precursor intensities were aggregated for each peptide to obtain peptide-level intensities. Peptides with any missing values were removed, and the remaining peptides were subjected to empirical Bayes-moderated $t$-statistics analysis between ligand-treated samples and control samples using the limma R package (v.3.58.1). To enable protein-level screening, each protein was represented by the peptide with the minimal $P$ value (two-sided empirical Bayes $t$-test) among all its quantified peptides; that is, representative peptides. The protein-level volcano plot was generated by using $-\log_{10}(P)$ and $\log_2$(fold change) of the representative peptide as the $y$ axis and $x$ axis, respectively. The empirical Bayes $t$-test $P$ value is used to rank the probability of a protein being recognized as a target hit. Unless

otherwise stated, only proteins displaying an increased stability by ligand treatment (that is, $\log_2$(fold change) < 0) are considered for candidate target protein ranking.

For PISA and TPP sample analysis, the protein group matrix outputs from DIA-NN are used for downstream statistical analysis. Proteins with any missing values in any of the four replicates are removed, and the remaining proteins are subjected to two-sided empirical Bayes-moderated $t$-statistic analysis between ligand-treated samples and control samples. The empirical Bayes $t$-test $P$ value (two-sided) is used to rank the probability of a protein being recognized as a target hit. For HT-PELSA, PISA and TPP, only proteins showing increased stability are considered potential targets. In HT-PELSA, this corresponds to reduced susceptibility to trypsin digestion (that is, $\log_2$(fold change) < 0), while in PISA, it indicates decreased precipitation and higher abundance in the supernatant (that is, $\log_2$(fold change) > 0).

For dose–response data analysis in HT-PELSA, the peptide-level intensities are calculated based on their respective precursors as above. Peptides are required to be quantified in at least three replicates of the dose–response experiment. For each replicate, the ratio (treated vs control) is calculated by dividing the peptide intensity at each ligand concentration by the control group's intensity. Before fitting dose–response curves, peptides are pre-filtered based on a minimum 30% stabilization or destabilization at the highest ligand concentration. Additionally, for stabilized peptides, the ratio at the second-highest concentration is required to be below one; for destabilized peptides, it is required to be above one. The ratio of peptide intensity at each ligand concentration was fitted to the ligand concentrations using a four-parameter log-logistic model in R (drc package, v.3.0.1). The $pEC_{50}$ was extracted from the fitting model. The correlation ($R^2$) to a sigmoid trend of the ligand dose–response profile for each peptide was derived by performing a Pearson correlation analysis between the estimated values and the original values. For candidate target peptide determination, the $R^2$ of peptides is required to be above 0.9 in at least three replicates; the mean value of the $pEC_{50}$ values across different replicates and the coefficient of variation of the $pEC_{50}$ values are calculated; peptides with a coefficient of variation of $pEC_{50}$ > 0.25 are excluded. The peptide with the highest $pEC_{50}$ is assigned to the corresponding protein to obtain a protein-level $pEC_{50}$ for each protein.

### Structural analysis
To evaluate the spatial relationships between peptides and annotated ligand-binding sites within proteins, we retrieved UniProt metadata files (.json) and AlphaFold-predicted protein structures (.pdb) for each unique protein of interest. Binding site annotations were extracted from the UniProt feature table by selecting entries labeled as 'Binding site', and the central residue of each annotated region was used as its representative. Protein structures were parsed using Biopython's PDB-Parser, and per-residue AlphaFold confidence scores were obtained from the $B$-factors of Cα atoms. For each peptide mapped to a protein, we calculated the minimum Euclidean distance between the Cα atom of the binding site residue and all atoms of the peptide's start and end residues. The smaller of these distances is used as the distance between the peptide and the binding site. These distances reflect straight-line measurements in three-dimensional space based on the static atomic coordinates provided by the AlphaFold-predicted models.

### Protein–protein interaction analysis
The BioGRID database was used for protein–protein interaction analysis (v.BIOGRID-ALL-4.4.247.tab3; retrieved July 2025). The dataset was filtered to retain only interactions involving $E. coli$ proteins with a corresponding UniProt accession. For each protein category—stabilized, destabilized and unresponsive—we generated all possible theoretical, non-redundant protein pairs. We then calculated the proportion of these pairs that corresponded to known interactions

reported in BioGRID. Statistical significance was assessed using Fisher's exact test, with unresponsive–unresponsive pairs serving as the background.

### Source of the known targets databases
The list of 523 human protein kinases was obtained from KinHub (http://www.kinhub.org). Kinase domain information, the mouse kinase list, human transmembrane protein list and $E. coli$ known ATP-binding proteins were all derived from UniProt. All lists can be found in the Supplementary Data 10.

### Gene ontology analysis
Gene Ontology analysis for human cell lines and $E. coli$ was performed with the clusterProfiler[31] R package using the enrichGO and enricher functions. For Gene Ontology analysis, the background was set as all quantified proteins in each dataset.

### Reporting summary
Further information on research design is available in the Nature Portfolio Reporting Summary linked to this article.

## Data availability
All results and data are available online (https://github.com/nico-huttmann/HT-PELSA and https://apps.embl.de/htpelsaapp). All raw files, search parameters and search outputs were deposited to the ProteomeXchange Consortium through the PRIDE partner repository under dataset identifier PXD062869. The source data corresponding to the main figures and Extended Data figures are published alongside the study. Supplementary Data are available through Figshare at https://doi.org/10.6084/m9.figshare.30191833 (ref. 32). Source data are provided with this paper.

## Code availability
Code to reproduce the analysis and figures of this study, along with a separate, user-friendly R package to perform the analysis, is available on GitHub (https://github.com/nicohuttmann/HT-PELSA).

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

## Acknowledgements
We thank A. Brauer from the Zimmermann lab, E. de la Cueva Bueno and A. Grassi from Laboratory Animal Resources (LAR) at EMBL for providing the mouse tissues. We thank S. Scheu from the Proteomics Core Facility at EMBL for providing the $E. coli$ samples. We thank D. Papagiannidis and other members of the Savitski lab for insightful discussions. M.M.S. is supported by the Allen Distinguished Investigator award through the Paul G. Allen Frontiers Group. The funder had no role in study design, data collection and analysis, decision to publish or preparation of the paper.

## Author contributions
K.L., C.M.P. and M.M.S. designed the study. K.L. and C.M.P. developed the methodology and performed the experiments together with I.B. K.L., N.H., M.G.-R. and M.L.B. developed the data analysis strategies and analyzed the data. K.L., C.M.P. and J.S. set up the mass

spectrometry workflows. I.B. finalized the figures for publication. K.L., C.M.P. and M.M.S. drafted the paper with contributions from I.B., M.G.-R., N.H. and J.S. The paper was reviewed and edited by all authors. M.M.S. supervised the study.

## Funding

## Competing interests

The authors declare no competing interests.

## Additional information

**Extended data** is available for this paper at https://doi.org/10.1038/s41594-025-01699-y.

**Correspondence and requests for materials** should be addressed to Mikhail M. Savitski.

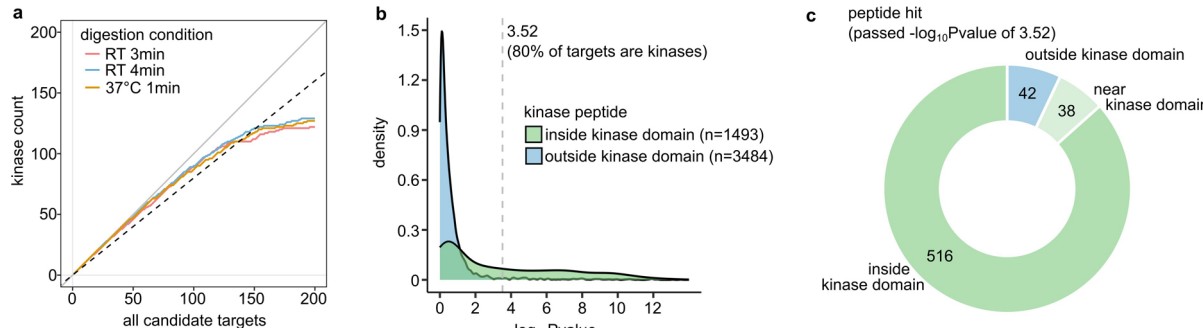

**Extended Data Fig. 1 | Benchmarking HT-PELSA for target identification and binding region determination with staurosporine. a**, True positive rate evaluation for different digestion conditions in HT-PELSA for staurosporine target identification. The gray line (slope = 1) and black dashed line (slope = 0.8) represent scenarios where 100% and 80% of the candidate targets are kinase targets, respectively. **b**, Density plots showing −log$_{10}$P distributions of all quantified peptides belonging to kinases, with tryptic cleavage sites located within or outside the kinase domains in HT-PELSA (RT 4 min) staurosporine analyses. The dashed line indicates the -log$_{10}$P above which more than 80% of stabilized proteins are kinases. **c**, The doughnut chart shows the locations of kinase peptides meeting the −log$_{10}$P cutoff from (**b**) and a log$_2$FC (treated vs. control) < 0. "inside" denotes peptides with N- or C-termini within the kinase domain; "near" indicates N- or C-termini within 10 amino acids of the kinase domain; otherwise, peptides are classified as "outside.

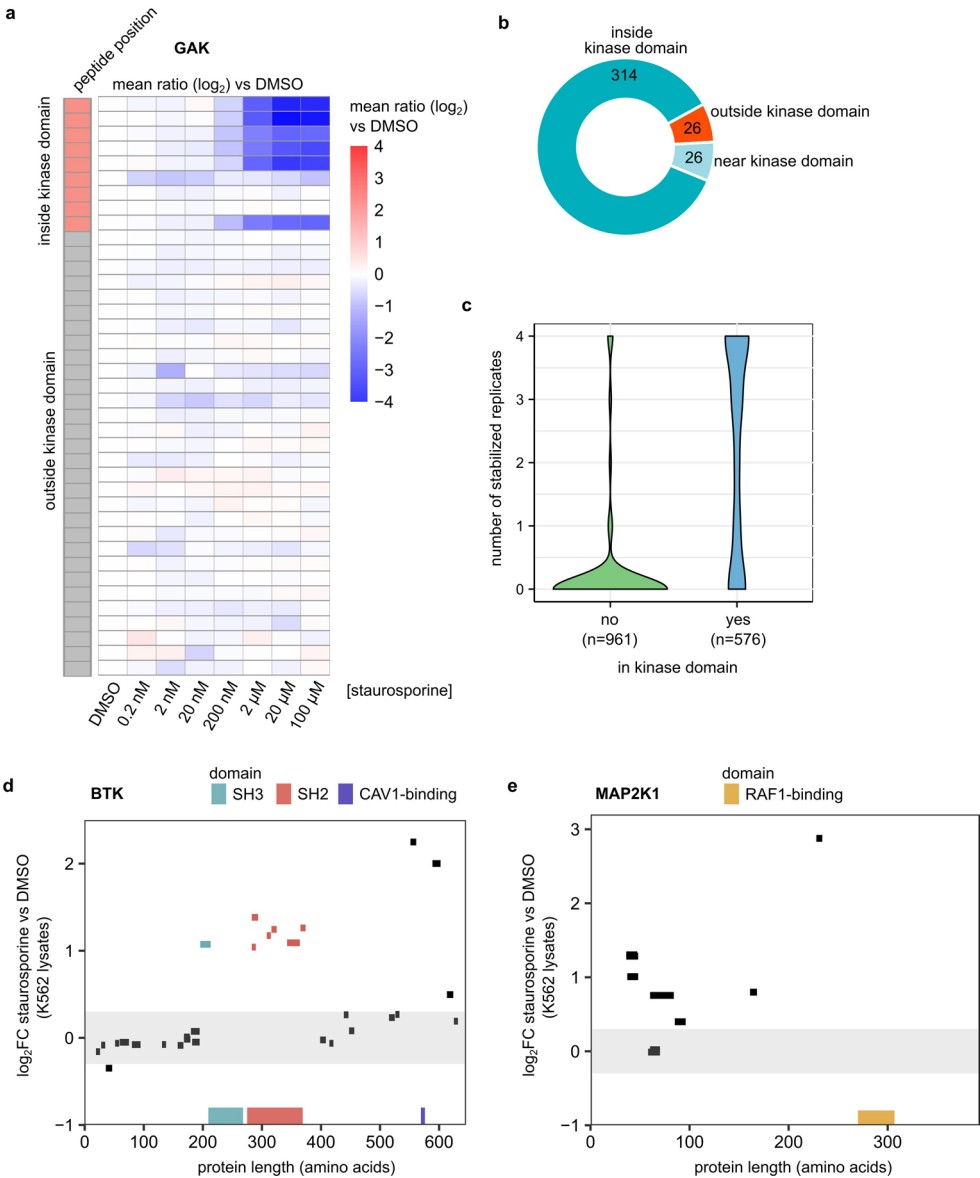

**Extended Data Fig. 2 | Dose-response HT-PELSA data reveals stabilized ligand-binding regions and destabilized protein-protein interacting interfaces.**
**a**, Heatmap representation of log$_2$ peptide fold changes of GAK with increasing staurosporine concentrations. The ratio represents the mean value across multiple replicates (at least three replicates are required). **b**, The doughnut chart shows the locations of kinase peptides meeting the criteria for dose dependent stabilization; at least 30% stabilization and dose-response curve fit (R$^2$) > 0.9 in at least three out of four replicates. "inside" denotes peptides with N- or C-termini within the kinase domain; "near" indicates N- or C-termini within 10 amino acids of the kinase domain; otherwise, peptides are classified as "outside". **c**, The distribution of the number of times peptides from hit kinases (those containing

at least one dose-dependent stabilized peptide, 93 hit kinases in total) inside or outside the kinase domain are detected as stabilized across the four replicates. **d**, Local stability profiles of BTK following 20 μM staurosporine treatment. SH3 and SH2 are known protein-protein interaction domains. The kinase domain of BTK spans from 402 to 655 and contains a unique protein-binding motif, CAV1-binding motif. **e**, Local stability profiles of MAP2K1 following 20 μM staurosporine treatment. The kinase domain of MAP2K1 spans from 68 to 361 and contains a unique protein-binding motif, RAF1-binding motif. **d** and **e**, the upper and lower boundaries of the gray-shaded area represent log$_2$FCs of 0.3 and −0.3, respectively. The x axis represents the protein sequence from the N-terminal to the C-terminal with known protein-protein interaction domains labelled.

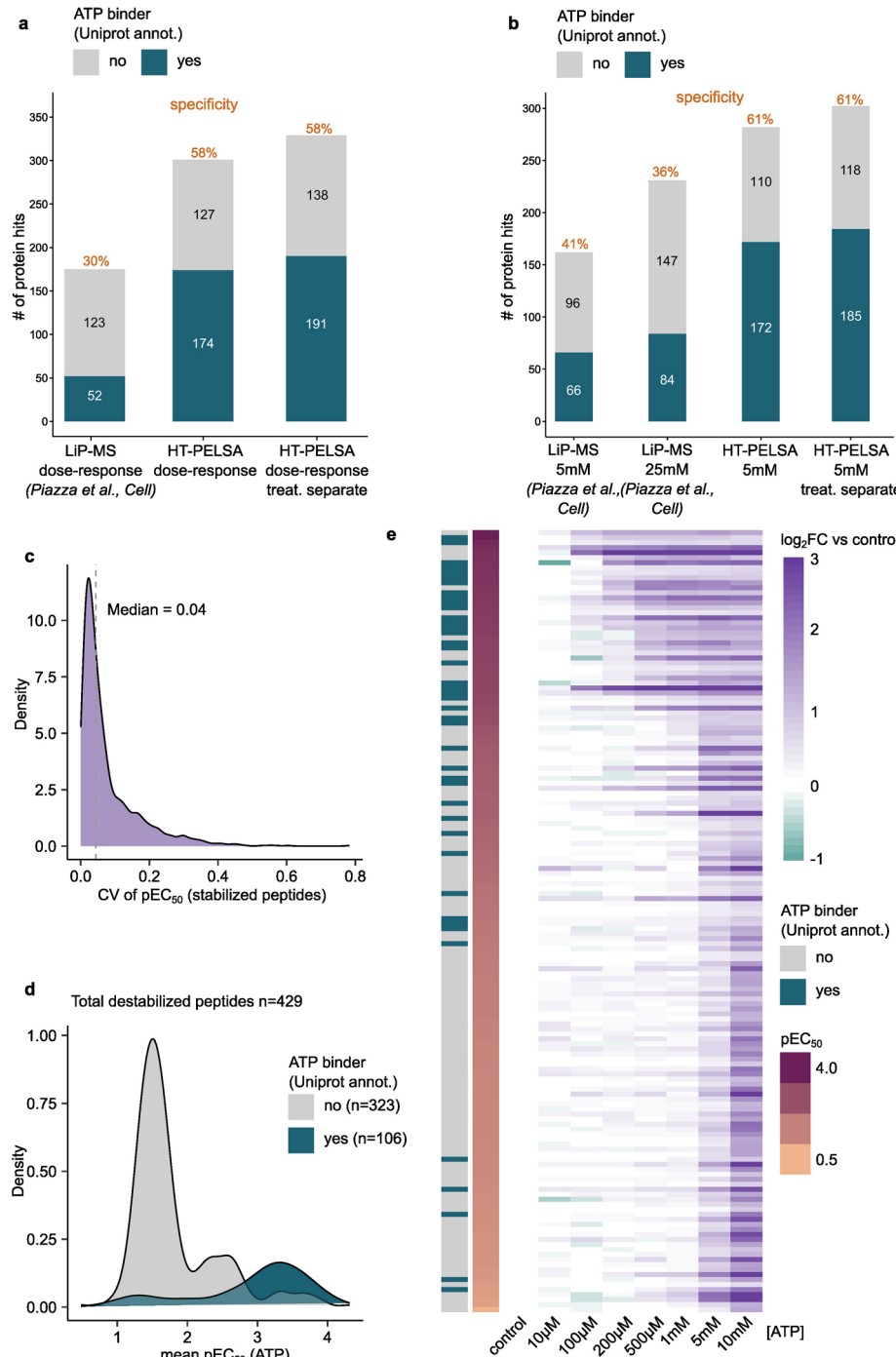

**Extended Data Fig. 3 | Characterization of ATP-binding proteins identified by HT-PELSA in Escherichia coli lysates. a**, Comparison of the total number of identified protein targets and Uniprot-annotated ATP binders in a published LiP-MS dataset[12] and HT-PELSA (with treatment performed before aliquoting lysate or treatment performed separately for each replicate) based on their respective ATP dose-response data. **b**, Comparison of the total number of identified protein targets and UniProt-annotated ATP-binding proteins at single ATP concentrations: 5 mM and 25 mM in LiP-MS[12], and 5 mM in HT-PELSA (with treatment performed before aliquoting lysate or treatment performed separately for each replicate). **c**, Distribution of the coefficient of variation of $pEC_{50}$ values across at least three HT-PELSA replicates for all peptides exhibiting dose-dependent stabilization by ATP in HT-PELSA. Peptides were considered

dose-dependently stabilized if they showed at least a 30% increase in resistance towards trypsin at the highest ATP concentration, with an $R^2 > 0.9$ in at least three replicates. **d**, $pEC_{50}$ distributions of all peptides exhibiting dose-dependent destabilization by ATP in HT-PELSA, grouped by whether they originate from UniProt-annotated ATP-binding proteins or not. Peptides were considered dose-dependently destabilized if they showed at least a 30% decrease in resistance towards trypsin at the highest ATP concentration, with an $R^2 > 0.9$ in at least three replicates. **e**, Heatmap showing the $\log_2$ fold change (relative to control) of 156 proteins exhibiting dose-dependent destabilization by ATP. Proteins are annotated according to whether they are UniProt-annotated ATP-binding proteins.

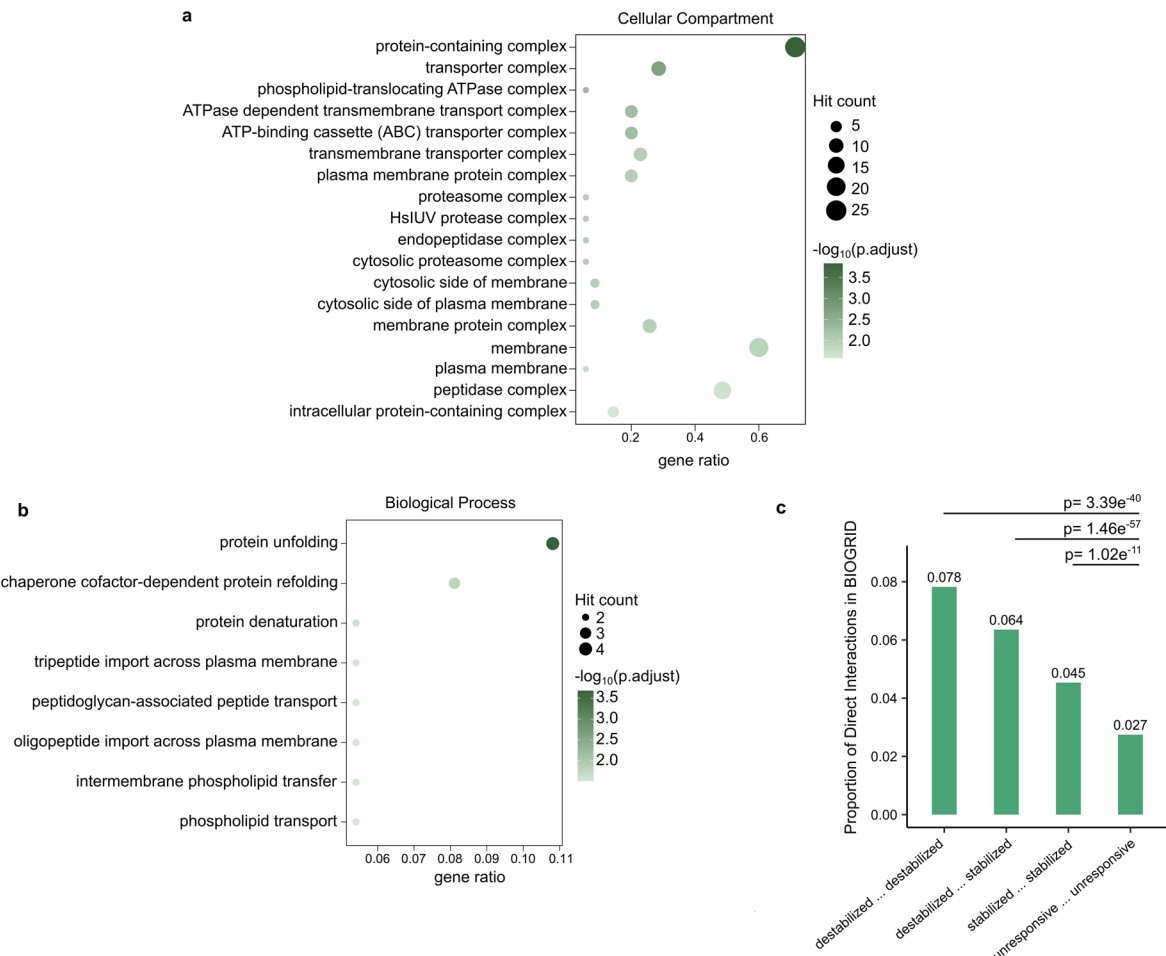

**Extended Data Fig. 4 | Characterization of destabilized proteins identified by HT-PELSA by ATP-treated Escherichia coli lysates. a**, GO cellular component analysis of the high-affinity ($pEC_{50} > 3$) ATP-destabilized proteins (39 proteins total). **b**, GO biological process analysis of the high-affinity ($pEC_{50} > 3$) ATP-destabilized proteins (39 proteins total). **c**, Proportion of protein-protein pairs with known interaction evidence, as reported in the BioGRID database (version BIOGRID-ALL-4.4.247.tab3; retrieved July 2025), for all possible pairwise combinations between proteins from different categories (Destabilized (39 proteins, $pEC_{50} > 3$), Stabilized (188 proteins, $pEC_{50} > 3$), and Unresponsive (2440 proteins). Statistical significance was assessed using two-sided Fisher's exact test (no adjustment), with unresponsive–unresponsive pairs serving as the background.

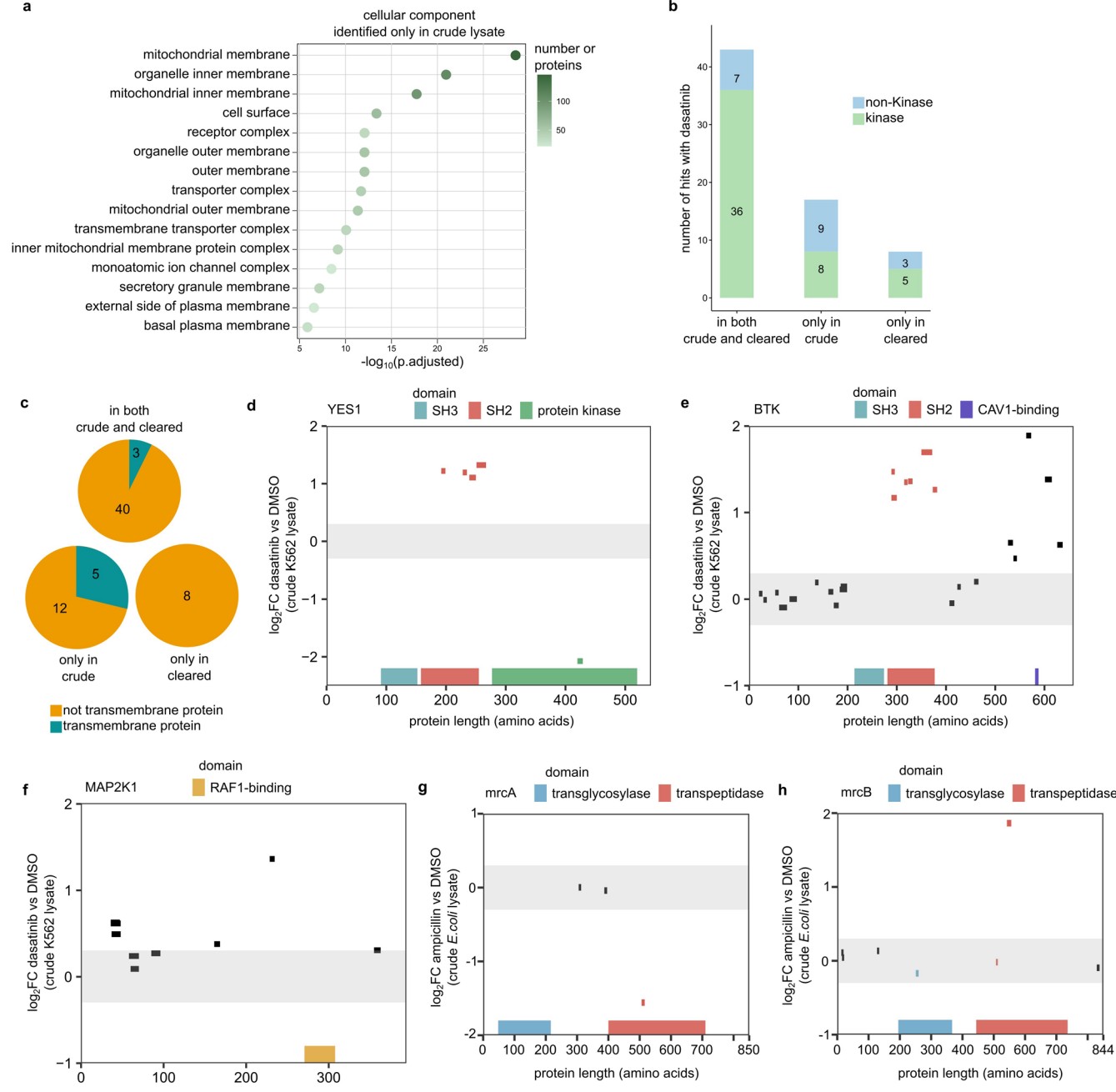

**Extended Data Fig. 5 | HT-PELSA in crude lysates for membrane target identification and binding region determination. a**, Gene ontology (cellular component) analysis of 556 proteins that are uniquely identified in crude lysates versus cleared lysates of K562 cells. **b**, Comparison of the numbers of candidate dasatinib-binding targets identified in crude and cleared K562 cell lysates, from left to right: targets identified in both crude and cleared lysates, only in the crude lysates, and only in the cleared lysates. **c**, The percentage of trans-membrane targets for each category defined in (**b**). **d**, Local stability profiles of YES1 following dasatinib treatment. SH3 and SH2 are known protein-protein interaction domains. **e**, Local stability profiles of BTK following dasatinib

treatment. SH3 and SH2 are known protein-protein interaction domains. The kinase domain of BTK spans from 402 to 655 and contains a unique protein-binding motif, CAV1-binding motif. **f**, Local stability profiles of MAP2K1 following dasatinib treatment. The kinase domain of MAP2K1 spans from 68 to 361 and contains a unique protein-binding motif, RAF1-binding motif. **g**, Local stability profiles of mrcA following ampicillin treatment. **h**, Local stability profiles of mrcB following ampicillin treatment. **d-h**, the upper and lower boundaries of the gray-shaded area represent $\log_2$FCs of 0.3 and −0.3, respectively. The x axis represents the protein sequence from the N-terminal to the C-terminal.

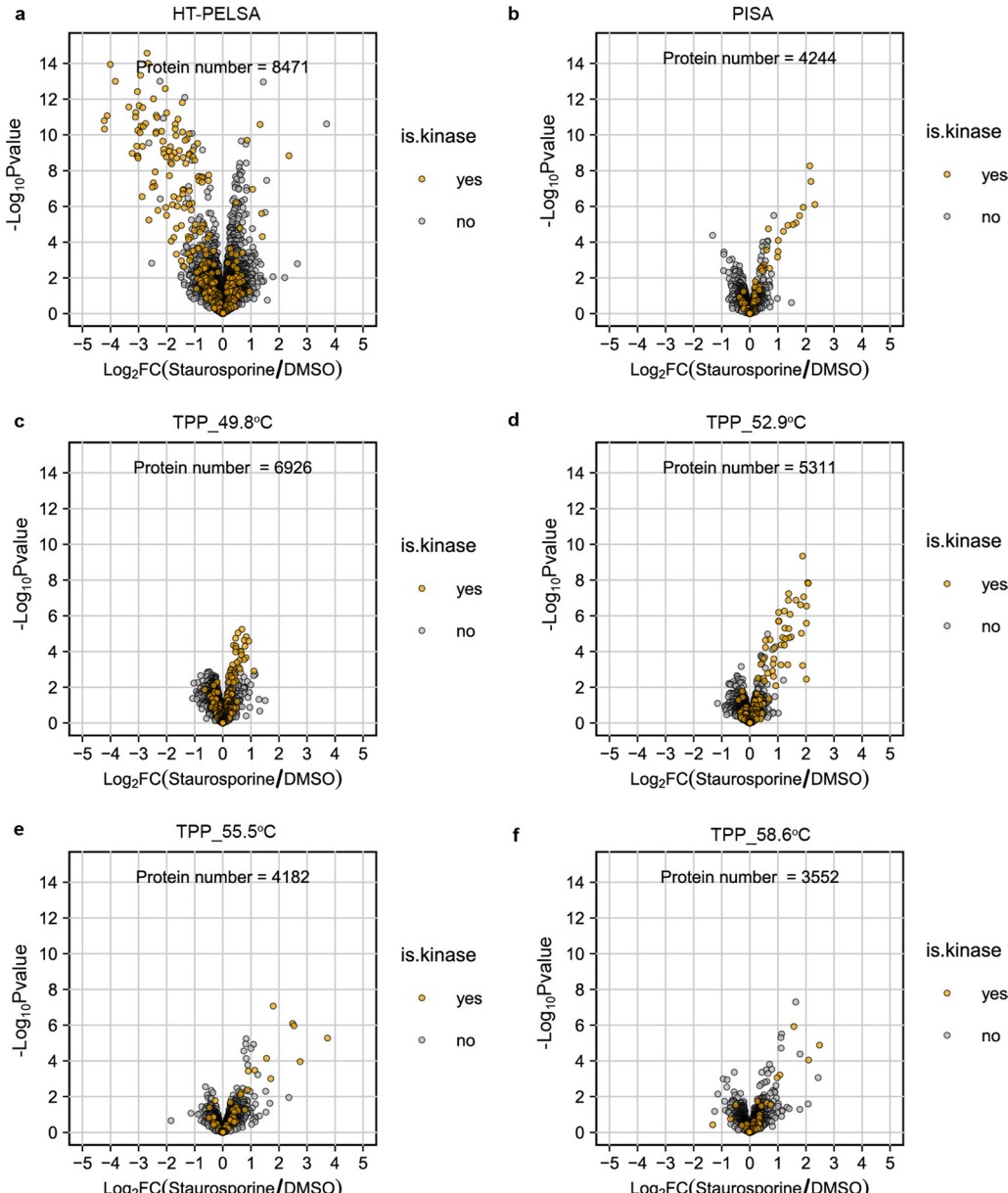

**Extended Data Fig. 6 | Comparison of HT-PELSA and existing methods for staurosporine target identification in crude liver lysates. a**, Volcano plot showing candidate staurosporine-binding targets identified by HT-PELSA in crude mouse liver lysates exposed to 20 μM staurosporine. **b**, Volcano plot showing candidate staurosporine-binding targets identified by PISA (52.9 °C, 55.5 °C, and 58.6 °C) in crude mouse liver lysates exposed to 20 μM staurosporine. **c**, Volcano plot showing candidate staurosporine-binding targets identified by TPP performed at 49.8 °C in crude mouse liver lysates exposed to 20 μM staurosporine. **d**, Volcano plot showing candidate staurosporine-

binding targets identified by TPP performed at 52.9 °C in crude mouse liver lysates exposed to 20 μM staurosporine. **e**, Volcano plot showing candidate staurosporine-binding targets identified by TPP performed at 55.5 °C in crude mouse liver lysates exposed to 20 μM staurosporine. **f**, Volcano plot showing candidate staurosporine-binding targets identified by TPP performed at 58.6 °C in crude mouse liver lysates exposed to 20 μM staurosporine. **a-f**, P values, a two-sided empirical Bayes t-test without adjustment (four treatment replicates, tissue lysates were aliquoted into separate portions, each treated individually with the compound or vehicle and subjected to independent HT-PELSA analysis).

# Reporting Summary

## Statistics

For all statistical analyses, confirm that the following items are present in the figure legend, table legend, main text, or Methods section.

| n/a | Confirmed | |
|---|---|---|
| ☐ | ☒ | The exact sample size (*n*) for each experimental group/condition, given as a discrete number and unit of measurement |
| ☐ | ☒ | A statement on whether measurements were taken from distinct samples or whether the same sample was measured repeatedly |
| ☐ | ☒ | The statistical test(s) used AND whether they are one- or two-sided<br>*Only common tests should be described solely by name; describe more complex techniques in the Methods section.* |
| ☒ | ☐ | A description of all covariates tested |
| ☐ | ☒ | A description of any assumptions or corrections, such as tests of normality and adjustment for multiple comparisons |
| ☐ | ☒ | A full description of the statistical parameters including central tendency (e.g. means) or other basic estimates (e.g. regression coefficient) AND variation (e.g. standard deviation) or associated estimates of uncertainty (e.g. confidence intervals) |
| ☐ | ☒ | For null hypothesis testing, the test statistic (e.g. *F*, *t*, *r*) with confidence intervals, effect sizes, degrees of freedom and *P* value noted<br>*Give P values as exact values whenever suitable.* |
| ☒ | ☐ | For Bayesian analysis, information on the choice of priors and Markov chain Monte Carlo settings |
| ☒ | ☐ | For hierarchical and complex designs, identification of the appropriate level for tests and full reporting of outcomes |
| ☐ | ☒ | Estimates of effect sizes (e.g. Cohen's *d*, Pearson's *r*), indicating how they were calculated |

*Our web collection on statistics for biologists contains articles on many of the points above.*

## Software and code

Policy information about availability of computer code

| | |
|---|---|
| Data collection | The mass spectrometers Orbitrap Exploris 480 and Astral were used for data acquisition. |
| Data analysis | Raw data was processed with DIA-NN (version 1.8.1). Protein structures are visualized with PyMOL (version 2.5.8). Data analysis was performed with R (4.3.3). Gene ontology analysis is performed with clusterProfiler ( 4.10.1). The code to reproduce the analysis and the figures can be found at Github (https://github.com/nicohuttmann/HT-PELSA). |

For manuscripts utilizing custom algorithms or software that are central to the research but not yet described in published literature, software must be made available to editors and reviewers. We strongly encourage code deposition in a community repository (e.g. GitHub). See the Nature Portfolio guidelines for submitting code & software for further information.

## Data

Policy information about availability of data

All manuscripts must include a data availability statement. This statement should provide the following information, where applicable:

- Accession codes, unique identifiers, or web links for publicly available datasets
- A description of any restrictions on data availability
- For clinical datasets or third party data, please ensure that the statement adheres to our policy

The raw mass spectrometry proteomics data, Fasta files, and DIA-NN output results have been deposited to the ProteomeXchange Consortium through the PRIDE partner repository with the dataset identifier PXD062869. Supplementary Data are available via Figshare at https://doi.org/10.6084/m9.figshare.30191833

Source Data for each figure and Supplementary datasets have been provided along with the paper. Protein structures are downloaded from Protein Data Bank or Alphafold.

## Research involving human participants, their data, or biological material

Policy information about studies with <u>human participants or human data</u>. See also policy information about <u>sex, gender (identity/presentation), and sexual orientation</u> and <u>race, ethnicity and racism</u>.

| | |
|---|---|
| Reporting on sex and gender | *Use the terms sex (biological attribute) and gender (shaped by social and cultural circumstances) carefully in order to avoid confusing both terms. Indicate if findings apply to only one sex or gender; describe whether sex and gender were considered in study design; whether sex and/or gender was determined based on self-reporting or assigned and methods used.*<br>*Provide in the source data disaggregated sex and gender data, where this information has been collected, and if consent has been obtained for sharing of individual-level data; provide overall numbers in this Reporting Summary. Please state if this information has not been collected.*<br>*Report sex- and gender-based analyses where performed, justify reasons for lack of sex- and gender-based analysis.* |
| Reporting on race, ethnicity, or other socially relevant groupings | *Please specify the socially constructed or socially relevant categorization variable(s) used in your manuscript and explain why they were used. Please note that such variables should not be used as proxies for other socially constructed/relevant variables (for example, race or ethnicity should not be used as a proxy for socioeconomic status).*<br>*Provide clear definitions of the relevant terms used, how they were provided (by the participants/respondents, the researchers, or third parties), and the method(s) used to classify people into the different categories (e.g. self-report, census or administrative data, social media data, etc.)*<br>*Please provide details about how you controlled for confounding variables in your analyses.* |
| Population characteristics | *Describe the covariate-relevant population characteristics of the human research participants (e.g. age, genotypic information, past and current diagnosis and treatment categories). If you filled out the behavioural & social sciences study design questions and have nothing to add here, write "See above."* |
| Recruitment | *Describe how participants were recruited. Outline any potential self-selection bias or other biases that may be present and how these are likely to impact results.* |
| Ethics oversight | *Identify the organization(s) that approved the study protocol.* |

Note that full information on the approval of the study protocol must also be provided in the manuscript.

# Field-specific reporting

Please select the one below that is the best fit for your research. If you are not sure, read the appropriate sections before making your selection.

☒ Life sciences ☐ Behavioural & social sciences ☐ Ecological, evolutionary & environmental sciences

For a reference copy of the document with all sections, see nature.com/documents/nr-reporting-summary-flat.pdf

# Life sciences study design

All studies must disclose on these points even when the disclosure is negative.

| | |
|---|---|
| Sample size | No statistical methods were used to predetermine sample size. To determine the significantly changed peptides or proteins, each experiment is performed with four replicates. Four replicates are sufficient to perform empirical Bayes moderated t-statistics to get the p values. As showed in the manuscript, after statistics analysis, the proteins with smallest p values are always known targets, confirming the effectiveness of using four replicates. The dose-response experiments were also performed with four replicates, and we required the target peptides fit the dose-response curves in at least three replicates; this allowed us to calculate the coefficient of variation (CV) of the pEC50 values derived from different replicates. By evaluating the CV and applying a cutoff, we were able to obtain consistent and reliable pEC50 values for each target peptide. |
| Data exclusions | For single concentration HT-PELSA experiment involving 8 samples, peptides with missing values in any of the replicates are removed. Since the number of the missing values increases with the number of sample, in dose-response HT-PELSA experiments involving 32 samples, we require peptides be quantified in at least three replicates and peptides quantified in less than three replicates, are removed. |
| Replication | Conclusions were drawn from reproducible effects in all replicates of the datasets. All experiments are performed using four replicates. |
| Randomization | The protein samples used for different treatments are from the same cell or tissue lysates. The lysates were thoroughly mixed and evenly distributed into separate tubes for vehicle or ligand treatment, ensuring no bias during distribution. |
| Blinding | This is not relevant to the study. After incubation with ligand or vehicle, the lysates are distributed into a 96-well plate; from that point onward, all samples are processed under identical conditions and for the same duration within the 96-well plate. |

# Reporting for specific materials, systems and methods

We require information from authors about some types of materials, experimental systems and methods used in many studies. Here, indicate whether each material, system or method listed is relevant to your study. If you are not sure if a list item applies to your research, read the appropriate section before selecting a response.

## Materials & experimental systems

| n/a | Involved in the study |
|-----|----------------------|
| ☒ | Antibodies |
| ☐ | ☒ Eukaryotic cell lines |
| ☒ | Palaeontology and archaeology |
| ☐ | ☒ Animals and other organisms |
| ☒ | Clinical data |
| ☒ | Dual use research of concern |
| ☒ | Plants |

## Methods

| n/a | Involved in the study |
|-----|----------------------|
| ☒ | ChIP-seq |
| ☒ | Flow cytometry |
| ☒ | MRI-based neuroimaging |

## Eukaryotic cell lines

Policy information about cell lines and Sex and Gender in Research

| | |
|---|---|
| Cell line source(s) | K562 cells are acquired from American Type Culture Collection (ATCC, CCL-243). |
| Authentication | Cells were authenticated by STR profiling by vendors. |
| Mycoplasma contamination | No mycoplasma contamination was discovered during the cell culture. |
| Commonly misidentified lines (See ICLAC register) | No misidentified cell lines have been used in this study. |

## Animals and other research organisms

Policy information about studies involving animals; ARRIVE guidelines recommended for reporting animal research, and Sex and Gender in Research

| | |
|---|---|
| Laboratory animals | The liver sample was collected from a 2-month-old C57BL/6 male mouse, and the heart sample from a 4-month-old C57BL/6J male mouse. |
| Wild animals | The study did not involve wild animals. |
| Reporting on sex | Sex-based analysis was not performed, as we believe that sex may not significantly impact the proteome of non-reproductive organs, such as the liver and heart. |
| Field-collected samples | The study did not involve sample collected from the field. |
| Ethics oversight | All animal care and procedures were conducted in line with EMBL regulations and guidelines for the use of animals in research and were reviewed and approved by the Institutional Animal Care and Use Committee (IACUC). All mouse experiments were performed using approved protocols by the European Molecular Biology Laboratory (EMBL) ethics committee (license 21-002_HD_MZ). |

Note that full information on the approval of the study protocol must also be provided in the manuscript.

## Plants

| | |
|---|---|
| Seed stocks | Report on the source of all seed stocks or other plant material used. If applicable, state the seed stock centre and catalogue number. If plant specimens were collected from the field, describe the collection location, date and sampling procedures. |
| Novel plant genotypes | Describe the methods by which all novel plant genotypes were produced. This includes those generated by transgenic approaches, gene editing, chemical/radiation-based mutagenesis and hybridization. For transgenic lines, describe the transformation method, the number of independent lines analyzed and the generation upon which experiments were performed. For gene-edited lines, describe the editor used, the endogenous sequence targeted for editing, the targeting guide RNA sequence (if applicable) and how the editor was applied. |
| Authentication | Describe any authentication procedures for each seed stock used or novel genotype generated. Describe any experiments used to assess the effect of a mutation and, where applicable, how potential secondary effects (e.g. second site T-DNA insertions, mosiacism, off-target gene editing) were examined. |

