## [Peer Review File · Nature Structural & Molecular Biology]

High-throughput peptide-centric local stability assay extends protein-ligand identification to membrane proteins, tissues, and bacteria

Corresponding Author: Dr Mikhail Savitski

Version 0:

Decision Letter:

11th Jun 2025

Dear Dr. Savitski,

Thank you again for submitting your manuscript "High-throughput peptide-centric local stability assay extends protein-ligand identification to membrane proteins, tissues, and bacteria". I apologise for the delay in responding, which resulted from the difficulty in timely obtaining and discussing referee reports. Nevertheless, we now have comments (below) from the 3 reviewers who evaluated your paper. In light of these reports, we remain interested in your study and would like to see your response to the comments of the referees, in the form of a revised manuscript.

Y

ou will see that while the reviewers appreciate the improvements made in HT-PELSA and its potential utility to the community, they raise notable concerns that must be addressed in a revised manuscript. More specifically, Reviewer #1 notes the importance of including biological replicates and internally benchmarking HT-PELSA with in-house acquired results using a relative method (such as PISA/TPP). Furthermore, Reviewer #2 provides useful recommendations for certain further analyses to showcase the application of HT-PELSA in the identification of off-target effects and requests demonstration of the complementarity of HT-PELSA with TPP or the potential for their integrative use. We editorially agree with Reviewers #1 and #2 with respect to performing these further experiments and analyses to further boost the confidence in the manuscript's technical applicability, but we disagree with respect to the view that this manuscript does not fall within scope for NSMB, so we don't request you specifically do something on this front. Finally, all reviewers request clarifications and further discussions at points which we ask you to heed to increase the clarity and accessibility of the manuscript.

Please be sure to address/respond to all concerns of the referees in full in a point-by-point response and highlight all changes in the revised manuscript text file. If you have comments that are intended for editors only, please include those in a separate cover letter. We also ask that you please submit the revised version of the manuscript as a Technical Report as this is more in line with the reported findings and advance, and may increase the manuscript's accessibility and presented main figures.

We expect to see your revised manuscript within 3-4 months. If you cannot send it within this time, please contact us to discuss an extension; we would still consider your revision, provided that no similar work has been accepted for publication at NSMB or published elsewhere.

Reporting Summary:

- that unprocessed scans are clearly labelled and match the gels and western blots presented in figures. Please note that all key data shown in the main figures as cropped gels or blots should be presented in uncropped form, with molecular weight markers. While these data can be displayed in a relatively informal style, they must refer back to the relevant figures. These data should be submitted as source data with the last revision, prior to acceptance.
- that control panels for gels and western blots are appropriately described as loading on sample processing controls
- all images in the paper are checked for duplication of panels and for splicing of gel lanes.
- For any revision that includes light microscopy data, we ask our authors to please include a completed light microscopy reporting table [https://www.nature.com/documents/Light_microscopy_reporting_table.xlsx] to ensure the methods are described thoroughly. The table will be available to reviewers and ultimately published should the manuscript be accepted at the journal.

EXTENDED DATA FIGURES

Data availability: this journal strongly supports public availability of data. All data used in accepted papers should be available via a public data repository, or alternatively, as Supplementary Information. If data can only be shared on request, please explain why in your Data Availability Statement, and also in the correspondence with your editor. Please note that for some data types, deposition in a public repository is mandatory - more information on our data deposition policies and available repositories can be found below:
<https://www.nature.com/nature-research/editorial-policies/reporting-standards#availability-of-data>

Link Redacted

Note: This URL links to your confidential home page and associated information about manuscripts you may have

submitted, or that you are reviewing for us. If you wish to forward this email to co-authors, please delete the link to your homepage.

Sincerely,

Dimitris Typas
Senior Editor
Nature Structural & Molecular Biology
ORCID: 0000-0002-8737-1319

Reviewers' Comments:

Reviewer #1 (Remarks to the Author):

Authors describe a "high-throughput adaption"(HT) of the previous published "Peptide-centric local stability assay" also known as PELSA (Li et al, Nature methods 2024). The principle of PELSA is pretty cool and could be very useful for the analysis of protein-drug interactions and generally applicable for not only basic research, but applied industrial applications such as drug discovery. There are lot of methods out there for this, some are easier than others, often being more of an art than a science due to the large number of steps. Thus, optimizations such as high-throughput adaption of PELSA (HT-PELSA) presented in this paper would always be useful to the community. Overall, the HT-PELSA optimizations (other than using the latest generation mass spectrometer to get the best numbers,), especially the sample preparation steps are sound and will lead to similar or slightly better results compared to the original method published last year. Nonetheless, the paper seems out of the scope for the journal even though the method itself I'm sure is nice. Many of the results presented in this study are ad-hoc comparisons to previously published data, published by different groups, using whole different setups, which is a pity because the authors could've really performed a fair comparison.

The results and analysis generally are a bit brief and a bit superficial. For example, the structural analysis of DnaK and Gluc in Figure 2. The authors use AlphaFold predicted structures for the DnaK analysis, and highlight the identified tryptic peptides on the structure with different colors alongside labeling of proximity to the binding sites and inferring some mechanism this way. There are many experimentally solved 3D structures on PDB and Uniprot for DnaK, why not use those instead? Many of the structures of DnaK are bound to different targets. Surely that would be a better way to explain why the proteolytic cleavage by Trypsin is better ? Maybe even try to add the Trypsin structure and try something to explain why the differential effect of protease cleavage occurs at the structural level? Should protein-protein interactions not be considered to explain some of these effects? The authors don't really expand on these ideas and therefore the structural analysis really overlooks different nuances and context for these proteins. I get AlphaFold is a great tool, but there should be a bit more analysis here other than putting up the predicted structure and highlighting peptides and regions.

The authors claim reproducibility of the method but that is not fully tested. For example, from the materials and method section and description of the "HT-PELSA workflow", it is stated that after incubation is performed the sample is split into 4, so it is technical replicates. Would it not make sense at the very minimum just do the drug treatments initially with replicates ? Surely that will be proper reproducibility of the method which takes into account the variance of the drug treatment ? I think it would be good to really clarify that this is not tested in the text, because that really makes some of the comparisons to other studies a bit unfair. For example, in Figure 2C, the results of the authors as presented are compared to Piazza et al, Cell. A paper from 2018 using a much older generation of instrument, and whose authors actually performed experiments in biological replicates.

Another example is the comparison of the streamlined PISA method demonstrated by Batth et al., Nature communication 2024, in Figure 3F. The authors compare the identification of liver targets (kinase and non-kinase). The Batth et al experimental procedures are performed with almost different procedures entirely, different Staurosporine concentrations are used, different time incubation, different incubation temperatures. The HT-PELSA has higher number of kinase targets while Batth et al, has more non-kinase targets, and more overall targets. Is it not plausible that the experimental procedures utilized by Batth et al. activate many of the liver enzymes which quickly degrade Staurosporine, so the targets that are actually measured are a result of downstream degradation product ? Again, the authors of the streamlined PISA utilize experimental replicates including the drug incubation, and their samples were analyzed on an older generation of mass spectrometry instrument. Lastly, Batth et al. PISA experiments were performed on Rat liver extracts, whereas here the authors used mice, this really is an apple to orange comparison. It is perplexing, the authors are one of the pioneers of the Thermal protein profiling (TPP) method, so why not just perform the TPP/PISA experiment in parallel ? Surely the authors would be one of the best to do it ? I'm sure the HT-PELSA method is probably on par or maybe slightly better, but the authors not doing this comparison makes it a bit sus.

The authors could obviously redo the experiments with proper replicates and alongside a comparison to PISA, but I am not sure it would still fit within the scope of the journal? It is a method optimization paper and is lacking in the structural and molecular biology analysis. Lastly, I'm also curious as to how the authors handle the "crude lysate", I'm guessing it is extremely sticky and "goeey", how do the authors deal with that ? The authors could expand a bit more on some parts of the experimental methods for clarity.

Overall, the paper and the HT-PELSA method is cool, it is a nice optimization of the original PELSAs no doubt. But the presentation here seems rushed, and there are some serious issues with the experimental comparisons. Lastly, the paper does not really seem to be within the scope and aims of this journal (NSMB).

Reviewer #2 (Remarks to the Author):

The manuscript by Li et al. introduces improvements to the peptide-centric local stability assay (PELSA) that enables significantly higher sample throughput and the usage of crude lysate, improving the probing of ligand binding to membrane bound proteins. The authors demonstrate the performance of this new workflow, HT-PELSA, within multiple and varied use cases, kinase-staurosporine binding, ATP binding in *E. Coli*, dasatinib binding by membrane-bound proteins in K562 cells, and sunitinib binding in mouse liver. Across these use cases the authors demonstrate that the workflow provides reproducible results that yield meaningful information and that outperform both the original PELSA method and another method that can be used to probe protein-ligand interactions, thermal proteome profiling (TPP). Overall, the authors demonstrate that their new HT-PELSA workflow offers a valuable way for researchers to systematically probe protein-ligand interactions, and this manuscript should be very well received by the scientific community.

My comments to the authors mostly revolve around some additional computational analysis that could aid in the understanding of the full scope of both the potential strengths/benefits and limitations of HT-PELSA.

Major Comments

1. The authors provide the example of DnaK, which is stabilized at the ATP binding site and destabilized at the substrate binding state. This suggests that, at least for some proteins, HT-PELSA results may be usable to identify off-target (i.e., not for the ligand being tested) binding sites, such as binding sites for protein interactors. Indeed, in the discussion, the authors suggest that HT-PELSA could be expanded to protein-protein or protein-nucleic acid interactions. As a small proof of concept, there are two quick data analyses that could be performed. A) For destabilized proteins, use annotations from PFAM or some other sequence-level annotation database and see across any or all of your HT-PELSA experiments if destabilized proteins are enriched for general protein binding or nucleic acid binding domains. B) Using PPI networks from databases like BioGRID or STRING, see if destabilized proteins are significantly more likely to be connected to a stabilized protein. Alternatively, you could calculate the distance within the graph between a destabilized protein to the nearest stabilized protein and see if this distance is significantly shorter than for random proteins.

2. For the *E. coli* ampicillin, the authors compare results from HT-PELSA to results from TPP from both lysate and live cells. Firstly, are the 4 known targets identified by TPP in live cells just a subset of the 5 identified by HT-PELSA? It would be helpful to list which specific known targets are also detected by TPP. Secondly, while HT-PELSA clearly has many benefits, one advantage that TPP has is that it can probe protein-ligand interactions in live cells, while HT-PELSA requires lysate. Thus, one can imagine that HT-PELSA would have a higher chance of identifying interactions that occur only in cell lysates, but not in living cells, or at least do not occur in live cells under a particular biological condition. What is the correlation or how similar are the TPP live cell and HT-PELSA results? If these are fairly similar, and especially if the TPP results are largely just a subset of HT-PELSA results, is there potential for integrating TPP and HT-PELSA results to get both the peptide-level resolution of HT-PELSA and live cell interaction context specificity of TPP?

Minor Comments

1. It is unclear what cells were used for the kinase-staurosporine interactions study. Based on Figure 1, it would seem that K562 cells were used, but please directly state in the main text which cells were used for this experiment.

2. In the discussion section, the authors write "With higher sensitivity than alternative approaches - currently requiring 60 μ g of protein input per sample, with only 5% of the resulting peptides being injected for mass spectrometry analysis - the input could be further scaled down." I assume based on the usage of 96-well plates that HT-PELSA uses far less than 60 μ g of proteins, but it is not clear if an estimation of the amount of protein needed for HT-PELSA is given. Providing even a rough estimate would be valuable for gaining a better sense of the scale of the reduction of sample input requirement.

Reviewer #3 (Remarks to the Author):

Li et al present a high throughput and optimized version of the PELSA approach for determining ligand-protein interactions. The authors make some relatively straightforward improvements in several areas that collectively result in a high throughput and reproducible plate-based workflow with access to lower solubility proteins. In my view these technical advancements are very welcome. Global proteomics combined with biophysical methods to detect protein-ligand interactions have been around for some time but in general have required fairly complex workflows that are difficult to scale reproducibly. This strategy would seem to overcome many of these. The manuscript is concisely written and straightforward (appreciated by this reviewer). I think this strategy will be of substantial interest and should be published without much fuss. I try to ask a few clarifying questions below that are generally minor in nature.

- Fig1C - the authors show the response curves for protected peptides in GAK on ligand binding. This is a nice visualization showing the value of dose-response (now feasible with the HT method). However, I think it would be helpful to show all peptides for this protein, possibly highlighting stabilized vs non-stabilized peptides (according to the 30% threshold). Do

some peptides have some intermediate behaviour or are they cleanly partition into stabilized and not stabilized (or destabilized)? Some general comment on how the peptide level response curves look across a given protein, and for all proteins that are considered hits would be of interest. From the original paper I think I remember some distribution.

- In the dasatinib data (where they show the most significant peptides per protein in the volcano) the most significant peptide for the expected target is down along with some off targets, but there are also quite a number that are up. In the ATP example this is rationalized with respect to orthosteric vs allosteric sites - this is fascinating. Is there anything to say about this in the dasatinib example where many kinases (BTK, etc) have the most significant peptide up in the treated condition? One would guess that the off targets would also be orthosteric or is there any other explanation of why some proteins show stabilization vs destabilization. Is it because only the most significant peptide per protein is shown and there is bidirectionality within given proteins? While this was explored in the ATP example perhaps some further exploration of the dasatinib (could be supplement) would be instructive.

- I initially found the nomenclature a bit tricky to follow in that 'stabilized' refers to peptides which are protected against digestion on liganding (i.e. the MS signal for that peptide goes down in the treatment/control comparison and they appear on left side of the volcano), whereas destabilized means signal for that peptide goes up on treatment. The authors note this very briefly but perhaps could be explained a bit more verbosely for clarity.

- For the crude lysates and membrane protein coverage, was there any consideration of including some weak detergent that could improve solubility but preserve native state - i.e. some low concentration NP-40 or similar? I notice some other biophysical methods have integrated this over the course of their evolution. Could it help?

- The discussion is a little sparse. While direct comparisons with other biophysical methods were made in the first pelsa paper and not necessarily here, perhaps these authors would like offer some thoughts with respect to whether the apparently much improved practicality (was demonstrated here) of this method might affect its utility with respect to other global proteomics methods for ligand-protein detection in the biophysical category.

signed
Ben Collins

Version 1:

Decision Letter:

Our ref: NSMB-BC50966A

5th Sep 2025

Dear Dr. Savitski,

Thank you for submitting your revised manuscript "High-throughput peptide-centric local stability assay extends protein-ligand identification to membrane proteins, tissues, and bacteria" (NSMB-BC50966A). It has now been seen by the two of the original referees and their comments are below. We could not yet obtain a report by original Reviewer #2. However, we did consult with both remaining reviewers on the points raised by Reviewer #2 and they deem them addressed. In summary, the reviewers find that the paper has improved in revision and therefore we are happy to accept it in principle in Nature Structural & Molecular Biology, pending minor revisions to satisfy the referees' final requests and to comply with our editorial and formatting guidelines. We also intend to change the manuscript from a Brief Communication to a Technical Report as we both think that it will increase its discoverability and because that's what our guidelines stipulate for new methods and method improvements like HT-PELSA.

We are now performing detailed checks on your paper and will send you a checklist detailing our editorial and formatting requirements in about 1-2 weeks. Please do not upload the final materials and make any revisions until you receive this additional information from us.

Sincerely,

Dimitris Typas
Senior Editor
Nature Structural & Molecular Biology
ORCID: 0000-0002-8737-1319

Reviewer #1 (Remarks to the Author):

The authors have done a good job of addressing the points. Perhaps it would also be a good idea to put in the observation regarding crude lysate preparation in the methods section as in the rebuttal?

"Using multiple freeze/thaw cycles for cell lysis in PBS, at a low lysate concentration (1.25 mg/mL of protein) without detergent ensures that the released DNA is sufficiently sheared and present at low concentration. This prevents the formation of the sticky, viscous texture typically caused by long strands of intact or aggregated DNA released under harsher lysis conditions."

I think this would be very useful information to the readers.

Congrats on a nice paper!!

Reviewer #3 (Remarks to the Author):

Comprehensive revision addressing all concerns. Recommend accept.

Version 2:

Decision Letter:

29th Sep 2025

Dear Dr. Savitski,

We are now happy to accept your revised paper "High-throughput peptide-centric local stability assay extends protein-ligand identification to membrane proteins, tissues, and bacteria" for publication as a Technical Report in Nature Structural & Molecular Biology.

Your paper will be published online soon after we receive proof corrections and will appear in print in the next available issue. You can find out your date of online publication by contacting the production team shortly after sending your proof corrections.

Authors may need to take specific actions to achieve compliance with funder and institutional open access mandates. If your research is supported by a funder that requires immediate open access (e.g. according to [Plan S principles](https://www.springernature.com/gp/open-science/plan-s-compliance) or the [NIH public access policy](https://www.springernature.com/gp/open-science/us-federal-agency-compliance)) then you should select the gold OA route, and we will direct you to the compliant route where possible. Because authors warrant under our subscription licensing terms that they haven't committed to licensing any version of their article under a licence inconsistent with the terms of our agreement – including the applicable embargo period – publication under the subscription model isn't suitable for authors whose funders require no embargo.

Sincerely,

Dimitris Typas
Senior Editor
Nature Structural & Molecular Biology
ORCID: 0000-0002-8737-1319

We thank the reviewers for their time and constructive comments that helped us to improve the manuscript. A detailed response to all comments can be found below.

Reviewers' Comments:

Reviewer #1 (Remarks to the Author):

Authors describe a "high-throughput adaption"(HT) of the previous published "Peptide-centric local stability assay" also known as PELSA (Li et al, Nature methods 2024). The principle of PELSA is pretty cool and could be very useful for the analysis of protein-drug interactions and generally applicable for not only basic research, but applied industrial applications such as drug discovery. There are lot of methods out there for this, some are easier than others, often being more of an art than a science due to the large number of steps. Thus, optimizations such as high-throughput adaption of PELSA (HT-PELSA) presented in this paper would always be useful to the community. Overall, the HT-PELSA optimizations (other than using the latest generation mass spectrometer to get the best numbers,), especially the sample preparation steps are sound and will lead to similar or slightly better results compared to the original method published last year. Nonetheless, the paper seems out of the scope for the journal even though the method itself I'm sure is nice. Many of the results presented in this study are ad-hoc comparisons to previously published data, published by different groups, using whole different setups, which is a pity because the authors could've really performed a fair comparison.

Reply

We thank the reviewer for acknowledging the significance and value of our work for both basic research and industrial applications. In particular, the application to tissues and membrane proteins represents a major step up from the original protocol and in combination with the high-throughput sample preparation should have a major impact in the field. Below we extensively address the comments raised by the reviewer.

The results and analysis generally are a bit brief and a bit superficial. For example, the structural analysis of DnaK and GluaC in Figure 2. The authors use Alphafold predicted structures for the DnaK analysis, and highlight the identified tryptic peptides on the structure with different colors alongside labeling of proximity to the binding sites and inferring some mechanism this way. There are many experimentally solved 3D structures on PDB and Uniprot for DnaK, why not use those instead? Many of the structures of DnaK are bound to different targets. Surely that would be a better way to explain why the proteolysis cleavage by Trypsin is better ? Maybe even try to add the Trypsin structure and try something to explain why the differential effect of protease cleavage occurs at the structural level? Should protein-protein interactions not be considered to explain some of these effects? The authors don't really expand on these ideas and therefore the structural analysis really overlooks different nuances and context for these proteins. I get AlphaFold is a great tool, but there should be a bit more analysis here other than putting up the predicted structure and highlighting peptides and regions.

Reply

Thank you for pointing this out. We completely agree with the reviewer that experimentally solved 3D structure may be more context-relevant, for example, revealing protein dimers or

showing the protein as part of a large complex. That is why we always use PDB structures whenever they are available; AlphaFold predictions are only used when no experimental structures exist for the protein.

For DnaK, we did use the PDB structure (PDB: 4B9Q). We used this DnaK structure because it happens to be in complex with ATP and shows the protein conformation after it binds ATP. At the same time it is a full-length structure, which enables showing all the changed peptides on the proteins.

Protein-protein interactions were considered here to explain the destabilization of DnaK. According to Uniprot: "*Unfolded proteins bind initially to DnaJ; upon interaction with the DnaJ-bound protein, DnaK hydrolyzes its bound ATP, resulting in the formation of a stable complex. GrpE releases ADP from DnaK; ATP binding to DnaK triggers the release of the substrate protein, thus completing the reaction cycle.*" Based on this mechanism, we propose that the increased trypsin accessibility observed in the substrate-binding domain may be attributed to the release of the substrate protein. We clarified the text in the manuscript as follows:

"This supports the notion that substrate release is triggered by ATP binding at sufficiently high concentrations, making the non-bound substrate-binding domain more accessible to trypsin."

We appreciate the suggestion to compare the conformations of DnaK under different states to see if our method could capture such conformation changes. But we couldn't find any available structures of DnaK in its substrate-bound state.

The authors claim reproducibility of the method but that is not fully tested. For example, from the materials and method section and description of the "HT-PELSA workflow", it is stated that after incubation is performed the sample is split into 4, so it is technical replicates. Would it not make sense at the very minimum just do the drug treatments initially with replicates? Surely that will be proper reproducibility of the method which takes into account the variance of the drug treatment? I think it would be good to really clarify that this is not tested in the text, because that really makes some of the comparisons to other studies a bit unfair. For example, in Figure 2C, the results of the authors as presented are compared to Piazza et al, Cell. A paper from 2018 using a much older generation of instrument, and whose authors actually performed experiments in biological replicates.

Reply

Thank you for pointing this out. The replicates here were performed treating the lysate with a drug and subsequently aliquoting and performing PELSA on the different aliquots, thus obtaining replicates of the PELSA assay itself. We have found before that this yields the same results as when performing the drug treatment separately.

As suggested by the reviewer, we have now added new liver-staurosporine and ATP-*E. coli* experiments to show that treating separate lysate aliquots with the compound produces equally good results as aliquoting after compound treatment. The results of these experiments are in Figure S3A and Figure S3B for ATP, and in Figure 3G (shown in the reply to the next point of the reviewer).

Added in the manuscript:

“In addition to treating lysates with ATP or vehicle and then aliquoting into four replicates for separate HT-PELSA analyses, we also tested the performance of HT-PELSA when treating four separate lysate aliquots with ATP or vehicle. Our results showed that separate treatment replicates produce equally good results as when treatment is performed before aliquoting (191 unprot-annotated ATP-binders and 58% specificity for dose-response result; 185 unprot-annotated ATP-binders and 61% specificity for single concentration 5 mM result) (Figure S3A and S3B, Supplementary Dataset 4).”

We agree with the reviewer that the Cell paper from 2018 used an older generation instrument, and thus direct comparison of the number of the identified targets would not be fair. However, this was not our original intent. As stated in the manuscript, we referenced the 2018 Cell dataset because it represented the most comprehensive ATP-binding protein dataset available for *E. coli*.

Our dataset represents the most comprehensive collection to date, featuring over three times as many targets as the 2018 Cell dataset. Notably, it also includes a significantly higher proportion of known ATP-binding proteins (~60% vs. ~40%), demonstrating that HT-PELSA offers both greater specificity and deeper coverage of ligand–protein interactions. While improvements in instrumentation contribute to increased coverage, the enhanced specificity is inherent to the method itself. Furthermore, in a benchmarking experiment using staurosporine (Figure 1B) on a more complex human cell lysate, running the assay on slower instrumentation (ca 20 Hz vs. 150 Hz) resulted in only a 20% reduction in the number of known targets detected, underscoring the robustness of the approach even with older instrumentation.

We now make this point clear by explicitly stating:

“The combination of HT-PELSA with the Orbitrap Astral thus **yields a resource** that represents a substantial leap in coverage and specificity compared to the previous most comprehensive study systematically profiling protein-ATP interactions, using limited proteolysis-mass spectrometry (LiP-MS) (Figure 2C, Figure S3A, Supplementary Dataset 3).”

Another example is the comparison of the streamlined PISA method demonstrated by Batth et al., Nature communication 2024, in Figure 3F. The authors compare the identification of liver targets (kinase and non-kinase). The Batth et al experimental procedures are performed with almost different procedures entirely, different Staurosporine concentrations are used, different time incubation, different incubation temperatures. The HT-PELSA has higher number of kinase targets while Batth et al, has more non-kinase targets, and more overall targets. Is it not plausible that the experimental procedures utilized by Batth et al. activate many of the liver enzymes which quickly degrade Staurosporine, so the targets that are actually measured are a result of downstream degradation product ? Again, the authors of the streamlined PISA utilize experimental replicates including the drug incubation, and their samples were analyzed on an older generation of mass spectrometry instrument. Lastly, Batth et al. PISA experiments were performed on Rat liver extracts, whereas here the authors used mice, this really is an apple to orange comparison. It is perplexing, the authors are one of the pioneers of the Thermal protein profiling (TPP) method, so why not just perform the TPP/PISA experiment in parallel ? Surely the authors would be one of the best to do it ? I'm sure the HT-PELSA method is probably on par or maybe slightly better, but the authors not doing this comparison makes it a bit sus.

Reply

Thank you for these suggestions. We now have performed thermal proteome profiling both in the PISA-DIA format, as well as at individual temperatures in the common proteome melting range again using DIA. In addition we repeated the HT-PELSA experiment with individual staurosporine treatments. All the samples were analyzed on the same instrument using the same MS method, the same lysis buffer and the same mouse liver.

Briefly, we treated mouse liver lysates separately with 20 uM staurosporine or DMSO for four replicates. For TPP (performed at single temperature format as described in Savitski Science 2014 Figure 4, or Franken et al Nat Prot 2015 Figure 7, in this case using single concentrations instead of dose response) and PISA, the treated lysates were then divided into 12 different temperature conditions and heated for 3 minutes (37, 44, 46.9, 49.8, 52.9, 55.5, 58.6, 62, and 66 °C for TPP; 52.9, 55.5, and 58.6 °C for PISA, the same temperatures as used in the Batths paper). All samples were analyzed on Astral using DIA.

After removing proteins or precursors with missing values, we quantified 4244 proteins for the PISA compared to 8427 proteins for the HT-PELSA sample (treatment replicates). As a reference, we also checked the PISA dataset (Batth et al, 2024, supplemental dataset 2) and ~3500 proteins were quantified across four replicates for Rat liver, which confirms the adequate proteome coverage of our PISA dataset. The smaller number of proteins detected in PISA compared to HT-PELSA is likely due to the greater complexity of tissue lysates: abundant extracellular-matrix and other connective-tissue components can keep proteins insoluble and make them easy to precipitate (an essential step in TPP). The volcano plots and specificity curves clearly show that while the top hits in PISA are kinases, the number of kinases identified at 70% specificity is significantly higher for HT-PELSA (106 vs. 11 kinase hits). It is also clear that the new HT-PELSA results with separate compound treatment are of the same quality as the original HT-PELSA data (106 vs. 90 kinase hits).

For both HT-PELSA, PISA and TPP, only proteins showing increased stability are considered potential targets. In HT-PELSA, this corresponds to reduced susceptibility to trypsin digestion (i.e., $\log_2FC < 0$), while in PISA, it indicates decreased precipitation and higher abundance in the supernatant (i.e., $\log_2FC > 0$).

Analysis of individual temperatures revealed that the performances of TPP is best at 52.9°C, identifying most kinase hits, (42 kinases hits), at 70% specificity (in line with previous work Savitski et al Science Figure 4 (pan kinase inhibitor analyzed at 53°C) and Ruan et al, Anal Chem, 2022, Figure 1E), followed by 49.8°C where the proteome and kinase coverage is higher, but the kinase fold changes are much smaller since most of them have not melted at that temperature (36 kinase hits at 70% specificity) and finally the worst results in terms of proteome coverage and kinase hit identification are obtained at 55.5°C and 58.6°C (4 and 0 kinases at 70% specificity). The results at other temperature points are not shown, as kinases had either not yet begun to precipitate or were almost fully precipitated, resulting in the identification of few kinase targets.

Taken together, this shows that TPP can identify kinases with good specificity in tissue but more than two fold fewer than HT-PELSA. It also highlights that care needs to be taken when performing experiments in the PISA format since including temperatures at which the proteome coverage is low and the response to treatment is minimal into the pool will adversely affect the results even when few temperatures are pooled.

We have now removed the figure comparing the results of the Bath et al rat PISA to mouse HT-PELSA. Instead, we incorporated the above figures, where HT-PELSA, PISA, and TPP at different temperatures performed on mouse liver are compared, as Figure 3G and Figure S6 in the manuscript.

The following condensed version of the text above is added to the manuscript:

“106 kinase targets are obtained when treating the crude lysate aliquots separately with staurosporine and performing HT-PELSA (Figure 3G, Figure S6A, Supplementary Dataset 8), again showing that whether treatment is performed together or separately does not significantly impact the performance of HT-PELSA.

To enable a parallel comparison of HT-PELSA with existing methods for target identification in crude tissue lysates, we also performed PISA and TPP to identify staurosporine targets in the mouse liver lysates with the same experimental settings as in HT-PELSA (treatment is performed separately for different replicates). PISA samples were obtained by pooling the supernatants after heat treatment at 52.9°C, 55.5°C, 58.6°C, as suggested by the PISA tissue study¹⁹. 4244 proteins were quantified for the PISA. As a reference, ~3500 proteins were quantified for Rat liver in the previously reported PISA dataset¹⁹, confirming the adequate proteome coverage of our PISA dataset. The smaller number of proteins detected in PISA compared to HT-PELSA (~8400 proteins) is likely due to the greater complexity of tissue lysates: abundant extracellular-matrix and other connective-tissue components can keep proteins insoluble and make them easy to precipitate (an essential step in TPP). The volcano

plots and specificity curves (Figure 3G and Figure S6B) clearly show that while the top hits in PISA are kinases, the number of kinases identified at 70% specificity is significantly higher for HT-PELSA (106 vs. 11 kinase hits) (Supplementary Dataset 8).

Analysis of individual temperatures revealed that the performances of TPP is best at 52.9°C (Figure 3G and Figure S6C, Supplementary Dataset 8), showing the best specificity and highest kinase coverage in line with previous work^{3,20}, (42 kinase hits at 70% specificity), followed by 49.8°C where the proteome and kinase coverage is higher, but the kinase fold changes are much smaller since most of them have not melted at that temperature, (36 kinase hits at 70% specificity), (Figure 3G and S6D, Supplementary Dataset 8), and finally the worst result in terms of proteome coverage and fold changes of kinases is obtained at 55.5°C and 58.6°C, (4 and 0 kinases at 70% specificity), (Figure 3G and S6E,F, Supplementary Dataset 8). Taken together, this shows that HT-PELSA is a very sensitive method for identifying drug targets in the complex tissue lysates. TPP can identify drug targets with good specificity in tissue but two-fold less than HT-PELSA. It also highlights that care needs to be taken when performing experiments in the PISA format since including temperatures at which the proteome coverage is low and the response to treatment is minimal into the pool will adversely affect the results even when few temperatures are pooled.”

The authors could obviously redo the experiments with proper replicates and alongside a comparison to PISA, but I am not sure it would still fit within the scope of the journal? It is a method optimization paper and is lacking in the structural and molecular biology analysis. Lastly, I'm also curious as to how the authors handle the “crude lysate”, I'm guessing it is extremely sticky and “goosey”, how do the authors deal with that ? The authors could expand a bit more on some parts of the experimental methods for clarity.

Reply

Using multiple freeze/thaw cycles for cell lysis in PBS, at a low lysate concentration (1.25 mg/mL of protein) without detergent ensures that the released DNA is sufficiently sheared and present at low concentration. This prevents the formation of the sticky, viscous texture typically caused by long strands of intact or aggregated DNA released under harsher lysis conditions.

In contrast to TPP/PISA experiments that require a high protein concentration to enable proper protein precipitation, the protein concentration required for HT-PELSA is fairly low (1.25 mg/mL); the lysate is not sticky or “goosey” and instead is homogenous at that concentration. Also in HT-PELSA, proteins don't necessarily have to be solubilized in the lysates and as long as the lysate is homogenous, they can be evenly distributed for different treatments and replicates. We also found that the desalting column performs well even with crude lysates containing large cell debris. As suggested by the reviewer, we have expanded this part in the Method section.

“Although the crude *E. coli* lysates contained insoluble debris, they were homogeneous and could be evenly distributed for different treatments and replicates.”

“The cell debris can pass through the SEC desalting column and stays in the desalted lysates.”

Overall, the paper and the HT-PELSA method is cool, it is a nice optimization of the original PIELSA no doubt. But the presentation here seems rushed, and there are some serious issues with the experimental comparisons. Lastly, the paper does not really seem to be within the scope and aims of this journal (NSMB).

Reply

We hope the multiple new datasets discussed above address the reviewer's concerns about the experimental comparisons.

Reviewer #2 (Remarks to the Author):

The manuscript by Li et al. introduces improvements to the peptide-centric local stability assay (PELSA) that enables significantly higher sample throughput and the usage of crude lysate, improving the probing of ligand binding to membrane bound proteins. The authors demonstrate the performance of this new workflow, HT-PELSA, within multiple and varied use cases, kinase-staurosporine binding, ATP binding in *E. Coli*, dasatinib binding by membrane-bound proteins in K562 cells, and sunitinib binding in mouse liver. Across these use cases the authors demonstrate that the workflow provides reproducible results that yield meaningful information and that outperform both the original PELSA method and another method that can be used to probe protein-ligand interactions, thermal proteome profiling (TPP). Overall, the authors demonstrate that their new HT-PELSA workflow offers a valuable way for researchers to systematically probe protein-ligand interactions, and this manuscript should be very well received by the scientific community.

My comments to the authors mostly revolve around some additional computational analysis that could aid in the understanding of the full scope of both the potential strengths/benefits and limitations of HT-PELSA.

Major Comments

1. The authors provide the example of DnaK, which is stabilized at the ATP binding site and destabilized at the substrate binding state. This suggests that, at least for some proteins, HT-PELSA results may be usable to identify off-target (i.e., not for the ligand being tested) binding sites, such as binding sites for protein interactors. Indeed, in the discussion, the authors suggest that HT-PELSA could be expanded to protein-protein or protein-nucleic acid interactions. As a small proof of concept, there are two quick data analyses that could be performed. A) For destabilized proteins, use annotations from PFAM or some other sequence-level annotation database and see across any or all of your HT-PELSA experiments if destabilized proteins are enriched for general protein binding or nucleic acid binding domains. B) Using PPI networks from databases like BioGRID or STRING, see if destabilized proteins are significantly more likely to be connected to a stabilized protein. Alternatively, you could calculate the distance within the graph between a destabilized protein to the nearest stabilized protein and see if this distance is significantly shorter than for random proteins.

Reply

As suggested by the reviewer (suggestion A), we carried out enrichment analyses of the destabilized proteins using InterPro domain annotations to assess functional associations. As also observed in previous study (Ilaria et al, *Cell*, 2018, PMID: 29307493), high concentrations of ATP can chelate Mg^{2+} in lysates, leading to dissociation of the ribosomal complex and destabilization of many ribosomal proteins. To focus on physiologically relevant destabilization

events, we only consider high-affinity destabilized proteins (i.e., those with $pEC_{50} > 3$), resulting in a total of 39 proteins.

A substantial number of the high-affinity destabilized proteins are known ATP-binding proteins (23/39) and the domain enrichment analysis showed that typical ATP-binding domains (P-loop_NTPase and AAA+_ATPase domains) are enriched.

However, as shown in the original manuscript (Fig S2F and S2G before, Fig S4A and S4B in revised version), we observed notable enrichment for complex-related terms under the GO Cellular Component. Consistent with the example chaperone protein DnaK, we also see protein unfolding term significantly enriched for Biological Process categories, which are typically associated with binding to unfolding substrate proteins.

We also computed the shortest-path lengths between all protein pairs based on the BIOGRID database as suggested by the reviewer (suggestion B). This analysis did not reveal substantial differences across categories (average distance from 2.09 to 2.19, below). We think this may be because only proteins that directly bind the target proteins would be destabilized, when the target proteins bind the ligand. However, due to the highly interconnected nature of protein-protein interaction networks, particularly in comprehensive databases such as BIOGRID, most proteins are linked through short paths to many others. This high background indirect interaction may diminish the difference of the direct interactions.

To see if the destabilized proteins tend to have direct interactions between each other or with the stabilized proteins, we examined the proportion of direct interactions between protein pairs classified as destabilized, stabilized, or unresponsive, including all other possible pairwise combinations. The analysis revealed that pairs involving destabilized proteins exhibited the highest proportion of direct interactions. This supports the hypothesis that HT-PELSA captures proteins engaged in direct protein-protein interactions.

Statistical significance was assessed using Fisher's exact test, with unresponsive–unresponsive pairs serving as the background.

We now present the above results in Figure S4C and add the following text to the manuscript:

“Gene Ontology (GO) analysis revealed that the destabilized proteins show significant enrichment for complex-related Cellular Component terms and chaperone protein unfolding related Biological Process terms (Figure S4A,B). This suggests that PELSA can capture not only direct ligand targets but also changes in protein-protein interactions resulting from treatment. Supporting this, BioGRID data showed that destabilized proteins interact more frequently with each other or with stabilized proteins than pairs of unresponsive proteins (7.8% for destabilized protein pairs and 6.3% for destabilized-stabilized protein pairs compared to 2.7% of unresponsive proteins pairs) (Figure S4C).”

2. For the *E. coli* ampicillin, the authors compare results from HT-PELSA to results from TPP from both lysate and live cells. Firstly, are the 4 known targets identified by TPP in live cells just a subset of the 5 identified by HT-PELSA? It would be helpful to list which specific known targets are also detected by TPP. Secondly, while HT-PELSA clearly has many benefits, one advantage that TPP has is that it can probe protein-ligand interactions in live cells, while HT-PELSA requires lysate. Thus, one can imagine that HT-PELSA would have a higher chance of identifying interactions that occur only in cell lysates, but not in living cells, or at least do not occur in live cells under a particular biological condition. What is the correlation or how similar are the TPP live cell and HT-PELSA results? If these are fairly similar, and especially if the TPP results are largely just a subset of HT-PELSA results, is there potential for integrating TPP and HT-PELSA results to get both the peptide-level resolution of HT-PELSA and live cell interaction context specificity of TPP?

Reply

This is a good point! For in-vivo TPP, 130 targets were identified for ampicillin and these targets may include many downstream effects, resulting from cellular stress. Among the 130 targets, 4 are reported ampicillin-binding targets, including MrcA, FtsI, DacB, PbpG. These 4 targets can all be identified by HT-PELSA (Figure 3D). This indicates that HT-PELSA could capture most direct interactions that are identified by TPP. But as we treated ampicillin on the lysate level, HT-PELSA will miss the downstream events captured by in-vivo treatment which is a unique feature of TPP. Since HT-PELSA is more sensitive than TPP, this also gives good confidence that the targets that are only identified by TPP in cells and not by HT-PELSA in lysate are indeed most likely downstream effects and not direct binding events.

We add the following text to the results section:

“Notably, all targets identified by TPP—both in lysates and in vivo—are included among the five known targets identified by HT-PELSA; the additional targets identified in vivo by TPP (Figure 3E) should be due to downstream effects of inhibition of the main targets. The identified known ampicillin-binding proteins belong to three protein families: Penicillin-binding proteins, D-alanyl-D-alanine carboxypeptidases, and Peptidoglycan D,D-transpeptidases. Notably, the penicillin-binding proteins both *mrcA* and *mrcB* contain an N-terminal transglycosylase domain, annotated in UniProt as penicillin-insensitive, and a C-terminal transpeptidase domain, annotated as penicillin-sensitive. Consistently, our HT-PELSA results show that *mrcA* and *mrcB* are specifically perturbed at their C-terminal transpeptidase domains, instead of N-terminal domains (Figure S5G,H). We can conclude that HT-PELSA can very sensitively identify targets and binding regions in lysates while TPP can confirm that these interactions happen inside cells and additionally reveal downstream effects.”

We now also add the following text to the discussion regarding exciting future combinations of HT-PELSA with TPP:

“HT-PELSA is an in-lysate method; therefore, combining it with methods capable of systematic protein-ligand mapping in intact cells, such as TPP, will be highly impactful for drug discovery. The sensitive ligand-binding region detection enabled by HT-PELSA, when combined with in situ information on ligand binding from TPP, as well as the downstream effects that TPP can identify, will be very valuable for characterizing the mechanisms of action of small molecules.”

Minor Comments

1. It is unclear what cells were used for the kinase-staurosporine interactions study. Based on Figure 1, it would seem that K562 cells were used, but please directly state in the main text which cells were used for this experiment.

Reply

Thank you for pointing this out. We used K562 cells for all the human cell line HT-PELSA experiments. This information has been included in the main text.

“To validate the performance of HT-PELSA, we compared our high-throughput approach with the original method by identifying protein targets of staurosporine—a broad-spectrum kinase inhibitor commonly used as a benchmark for protein-ligand interaction studies—in K562 cell lysates.”

“To validate our methodology, we first determined the pEC50 values for kinase-staurosporine interactions in K562 cell lysates.”

2. In the discussion section, the authors write “With higher sensitivity than alternative approaches - currently requiring 60 µg of protein input per sample, with only 5% of the resulting peptides being injected for mass spectrometry analysis - the input could be further scaled down.” I assume based on the usage of 96-well plates that HT-PELSA uses far less than 60 µg of proteins, but it is not clear if an estimation of the amount of protein needed for HT-PELSA is given. Providing even a rough estimate would be valuable for gaining a better sense of the scale of the reduction of sample input requirement.

Reply

Thank you for pointing this out! For the starting amount per sample, we didn't optimize and instead followed the protocol from the original PELSA paper (Li et al, Nature Methods, 2025), using 50 uL lysates per sample with protein concentration of 1.2 mg/mL (based on BCA measurement), and we added 5 uL trypsin solution (5 mg/mL) for PELSA digestion. As only 4% of the resulting peptides are injected for mass spectrometry analysis with current protocol; the starting sample input could be in principle scaled down to 2.4 ug for future use, when working with precious material. However, this requires proper optimization—such as using smaller volumes, lower protein concentrations, and adjusting the amount of added trypsin.

We now add this as an outlook in the discussion section:

“The input could be further scaled down by decreasing the starting protein concentration or volume and optimizing digestion conditions accordingly.”

Reviewer #3 (Remarks to the Author):

Li et al present a high throughput and optimized version of the PELSA approach for determining ligand-protein interactions. The authors make some relatively straightforward improvements in several areas that collectively result in a high throughput and reproducible plate-based workflow with access to lower solubility proteins. In my view these technical advancements are very welcome. Global proteomics combined with biophysical methods to detect protein-ligand interactions have been around for some time but in general have required fairly complex workflows that are difficult to scale reproducibly. This strategy would seem to overcome many of these. The manuscript is concisely written and straightforward (appreciated by this reviewer). I think this strategy will be of substantial interest and should be published without much fuss. I try to ask a few clarifying questions below that are generally minor in nature.

- Fig1C - the authors show the response curves for protected peptides in GAK on ligand binding. This is a nice visualization showing the value of dose-response (now feasible with the HT method). However, I think it would be helpful to show all peptides for this protein, possibly

highlighting stabilized vs non-stabilized peptides (according to the 30% threshold). Do some peptides have some intermediate behaviour or are they clearly partition into stabilized and not stabilized (or destabilized)? Some general comment on how the peptide level response curves look across a given protein, and for all proteins that are considered hits would be of interest. From the original paper I think I remember some distribution.

Reply

Thank you for this comment. We performed four replicates of the dose-response experiment and included only peptides that were quantified in at least three replicates across the full range of compound concentrations for further analysis. For stabilized peptides, we applied a 30% stabilization threshold and required the dose-response curve fit (R^2) to exceed 0.9 in at least three replicates. Some peptides met these criteria in only one or two replicates and we guess these peptides may be the intermediate peptides the reviewer is referring to; while a portion of these may be true target peptides, our strict selection criteria filtered them out. We believe some of these peptides are changed randomly because of the fluctuation of quantification. Benefiting from the strict criteria, we could see that 93% kinase peptides showing dose-response stabilization are located within or near (within 10 amino acid distance) the kinase domains. But at the same time, we also observed that a number of peptides from the kinase domain met the criteria in fewer than three replicates. The figures have been included in the manuscript as Figure S2B and Figure S2C.

For the individual protein, GAK, consistent with the conclusion above, the peptides showing dose-dependent stabilization are all located within the kinase domain. There are also three peptides from the kinase domain that don't show dose-dependent stabilization.

We now add the heatmap below as Figure S2A

- In the dasatinib data (where they show their most significant peptides per protein in the volcano) the most significant peptide for the expected target is down along with some off targets, but there are also quite a number that are up. In the ATP example this is rationalized with respect to orthosteric vs allosteric sites - this is fascinating. Is there anything to say about this in the dasatinib example where many kinases (BTK, etc) have the most significant peptide up in the treated condition? One would guess that the off targets would also be orthosteric or is there any other explanation of why some proteins show stabilization vs destabilization. Is it because only the most significant peptide per protein is shown and there is bidirectionality within given proteins? While this was explored in the ATP example perhaps some further exploration of the dasatinib (could be supplement) would be instructive.

Reply

Thank you for raising this interesting question. In fact, some proteins show bidirectional effects, which are both stabilized and destabilized at the same time. For example, the change of kinase YES1 is similar to DnaK – it is stabilized at the kinase domain (i.e., direct binding domain) but destabilized at the SH2 domain (the protein-protein interacting domain).

Added to the main text: “Notably, the kinase YES1 was both stabilized at the kinase domain and destabilized at the SH2 domain (the protein-protein interacting domain) (Figure S5D).”

At the same time, we also found some kinases that are only destabilized by dasatinib treatment, also at the kinase domain, which is supposed to be the direct binding domain, for example, BTK, mentioned by the reviewer. Interestingly, we observed that within the kinase domain of BTK, there is a unique CAV1-binding motif, which is a protein interaction domain. Notably, BTK was destabilized exactly at the SH2 domain and in the proximity of the CAV1-binding region. Notably, the destabilization of BTK is also observed for another kinase inhibitor, staurosporine, also at the protein-interaction domains (SH3, SH2, and CAV1-binding motif).

The three protein-binding domains (from left to right: SH3, SH2, and CAV1-binding motif) are marked. The protein kinase domain of BTK spans from 402-655.

We also checked another significantly destabilized protein MAP2K1 and, interestingly, within the kinase domain of MAP2K1, there is also a unique RAF1-binding motif, which is a protein interaction motif. And the peptide showing strongest destabilization is the peptide closest to the RAF1-binding motif. Similar to BTK, this destabilization of MAP2K1 is reproducibly observed for both dasatinib and staurosporine at the same region.

The protein-binding motif (RAF1-binding motif) is marked in yellow and the kinase domain of MAP2K1 spans from 68 to 361.

Therefore, we suggest that it could be possible that the destabilization of the kinase, even at the kinase domain, could be caused by disrupted protein-protein interaction.

Added in main text:

- for staurosporine part: “Leveraging the peptide-level resolution of our HT-PELSA, we found that some destabilized kinases are destabilized precisely at or near well-characterized protein-binding motifs. For example, BTK shows destabilization exactly at the SH3 and SH2 domains, as well as around the CAV1-binding motif, (Figure S2D). In MAP2K1, the kinase domain contains a unique RAF1-binding motif, and the peptide showing the strongest destabilization is located closest to this motif (Figure S2E). These findings suggest that a possible explanation for kinase destabilization could be disrupted protein–protein interactions.”
- For dasatinib part: “Consistent with the staurosporine experiment, we also see destabilization of BTK and MAP2K1, with destabilized peptides occurring within or proximate to the annotated protein interaction domains (Figure S5E,F).”
- Added sentence in Discussion: “It is noteworthy that some direct targets are destabilized in their ligand binding domains in HT-PELSA experiments, this could potentially be explained by ligand binding competing away interacting proteins which could lead to destabilization if interactions were taking place at or close to the binding domain and occluding it. A mechanistic investigation of such cases will be a subject of future studies.”

- I initially found the nomenclature a bit tricky to follow in that 'stabilized' refers to peptides which are protected against digestion on liganding (i.e. the MS signal for that peptide goes down in the treatment/control comparison and they appear on left side of the volcano), whereas destabilized means signal for that peptide goes up on treatment. The authors note this very briefly but perhaps could be explained a bit more verbosely for clarity.

Reply

Thank you for the suggestion. We have added the clarification to the main text:

“Proteins are considered stabilized by the ligand if they become protected from digestion upon ligand binding. Accordingly, peptides that show decreased abundance in the treatment/control comparison are classified as stabilized peptides”.

- For the crude lysates and membrane protein coverage, was there any consideration of including some weak detergent that could improve solubility but preserve native state - i.e. some low concentration NP-40 or similar? I notice some other biophysical methods have integrated this over the course of their evolution. Could it help?

Reply

Thank you for this comment. In TPP indeed NP40 is often and ideally used post heating to extract the transmembrane proteins from the membrane before centrifugation or filtration. In that way the heat induced aggregation takes place in crude lysate or intact cells which is optimal for physiological relevance, as detergent can affect melting behavior (Reinhard et al, Nat Met 2015) and in some cases compound binding or affinity (Becher et al, Nat Chem Bio 2016). With PELSA this is not a concern since the peptides from the ligand binding domains will be released from the membrane embedded proteins in the crude lysate and there is no further need to extract the undigested protein remnants from the membrane.

- The discussion is a little sparse. While direct comparisons with other biophysical methods were made in the first pelsa paper and not necessarily here, perhaps these authors would like offer some thoughts with respect to whether the apparently much improved practicality (was demonstrated here) of this method might affect its utility with respect to other global proteomics methods for ligand-protein detection in the biophysical category.

Reply

Thank you for this comment. This is a very good suggestion and we now expand the discussion on the position of HT-PELSA with regard to some other prominent biophysical protein-ligand mapping methods.

“HT-PELSA is an in-lysate method; therefore, combining it with methods capable of systematic protein-ligand mapping in intact cells, such as TPP, will be highly impactful for drug discovery. The sensitive ligand-binding region detection enabled by HT-PELSA, when combined with in situ information on ligand binding from TPP, as well as the downstream effects that TPP can identify, will be very valuable for characterizing the mechanisms of action of small molecules.

We expect that the versatility with respect to source material and sensitivity of ligand binding detection of HT-PELSA will make it the method of choice for finding targets of uncharacterized compounds in lysate. However, if the goal is to assess compound selectivity within a specific enzyme family such as kinases, a kinase specific method like kinobeads will still offer higher sensitivity, but will generally not be able to detect non-kinase off-targets of kinase specific inhibitors.”

signed
Ben Collins